# Protein phosphatase 5 regulates titin phosphorylation and function at a sarcomere-associated mechanosensor complex in cardiomyocytes

Judith Krysiak[1], Andreas Unger[1,2], Lisa Beckendorf[1], Nazha Hamdani[1], Marion von Frieling-Salewsky[3], Margaret M. Redfield[4], Cris G. dos Remedios[5], Farah Sheikh[6], Ulrich Gergs [7], Peter Boknik[8] & Wolfgang A. Linke [3,9]

Serine/threonine protein phosphatase 5 (PP5) is ubiquitously expressed in eukaryotic cells; however, its function in cardiomyocytes is unknown. Under basal conditions, PP5 is auto-inhibited, but enzymatic activity rises upon binding of specific factors, such as the chaperone Hsp90. Here we show that PP5 binds and dephosphorylates the elastic N2B-unique sequence (N2Bus) of titin in cardiomyocytes. Using various binding and phosphorylation tests, cell-culture manipulation, and transgenic mouse hearts, we demonstrate that PP5 associates with N2Bus in vitro and in sarcomeres and is antagonistic to several protein kinases, which phosphorylate N2Bus and lower titin-based passive tension. PP5 is pathologically elevated and likely contributes to hypo-phosphorylation of N2Bus in failing human hearts. Furthermore, Hsp90-activated PP5 interacts with components of a sarcomeric, N2Bus-associated, mechanosensor complex, and blocks mitogen-activated protein-kinase signaling in this complex. Our work establishes PP5 as a compartmentalized, well-controlled phosphatase in cardiomyocytes, which regulates titin properties and kinase signaling at the myofilaments.

[1] Department of Cardiovascular Physiology, Ruhr University Bochum, MA 03/57, D-44780 Bochum, Germany. [2] Institute for Genetics of Heart Diseases, University Hospital Muenster, Domagkstraße 3, D-48149 Muenster, Germany. [3] Institute of Physiology II, University of Muenster, Robert-Koch-Staße 27b, D-48149 Muenster, Germany. [4] Division of Cardiovascular Diseases, Mayo Clinic, 200 1st St SW, Rochester, 55905 MN, USA. [5] Department of Anatomy & Histology, University of Sydney, Sydney, NSW 2006, Australia. [6] Department of Medicine, University of California-San Diego, 9500 Gilman Drive #0613-C, 92093 La Jolla, CA, USA. [7] Institute for Pharmacology and Toxicology, University of Halle-Wittenberg, Magdeburger Straße 4, 06112 Halle (Saale), Germany. [8] Institute for Pharmacology and Toxicology, University of Muenster, Domagkstraße 12, 48149 Muenster, Germany. [9] Deutsches Zentrum für Herz-Kreislaufforschung, Partner Site Goettingen, Robert-Koch-Str. 40, Goettingen, 37099, Germany. Correspondence and requests for materials should be addressed to W.A.L. (email: wlinke@uni-muenster.de)

During the lifetime of a beating heart, the cardiomyocytes must respond dynamically to a multitude of internal and external stresses. Such functional flexibility is supported at the level of the contractile units, the sarcomeres, by the expression of cardiac-specific isoforms of structural, contractile, and regulatory proteins. Some of them, such as cardiac troponin-I, myosin-binding protein-C, or titin, contain unique sequence motifs that can be phosphorylated and dephosphorylated by protein kinases and phosphatases, respectively. These selective biochemical events then help to quickly adjust the mechanical function of the cardiac sarcomere to altered physiological requirements, e.g., during exercise. In the diseased heart this fine-tuned mechanism can be disrupted. Whereas multiplex kinase signaling has been recognized as an important modifier of cardiac function at the level of sarcomeric proteins[1], much less is known about how this function is modulated by protein phosphatases[2].

Titin is a multifunctional protein giant, which determines the 'passive' elasticity of the sarcomere[3, 4] and also modulates active contractile properties[5–8]. Human titin encompasses up to ~ 36,000 amino acids encoded by the 364 exons of the *TTN* gene and probably is the protein with the most (potential) phosphorylation sites, but very few have been explored functionally[3]. Only one region in titin, termed N2B (encoded by exon 49 in mouse and human), is unique to the cardiac isoforms[9]. This region is located in the elastic (I-band) segment of the molecule and contains a 572-residue N2B-unique sequence (N2Bus), which is an important spring element[10]. Moreover, N2Bus is a hub for protein-protein interactions[3] and a major site for oxidation[11] and phosphorylation[12–15]. Several protein kinases (PKs) phosphorylate N2Bus, including PKA[12], PKG[13], the mitogen-activated protein kinase (MAPK) extracellular signal-regulated kinase 2 (ERK2, encoded by *MAPK1*)[14], and Ca$^{2+}$/calmodulin-dependent protein kinase IIδ (CaMKIIδ)[15]. Functionally, enhanced phosphorylation increases the distensibility of N2Bus and lessens the force needed to stretch this spring element, which lowers cardiomyocyte passive tension[3]. Whether this phosphorylation affects protein-protein interactions at N2Bus is unknown. Interestingly, N2Bus shows a phosphorylation deficit in human and experimental heart failure (HF), which contributes to increased myocardial passive stiffness, notably in HF with preserved ejection fraction (HFpEF)[3].

In the present study, we identified serine/threonine protein phosphatase 5 (PP5, encoded by *PPP5C*) as a novel interaction partner of N2Bus and a physiological antagonist to the PKs that phosphorylate N2Bus. PP5 is established as a ubiquitous enzyme in eukaryotic cells involved in multiple signaling pathways important for cell cycle progression, glucocorticoid receptor activation, DNA damage repair, transcriptional activation, or apoptosis[16, 17]. PP5 is unique among serine/threonine phosphatases as it contains a tetratricopeptide repeat (TPR) domain at the N-terminus, which binds to the C-terminal catalytic domain and blocks substrate access to the catalytic site, such that 'free' PP5 has low activity[18, 19]. However, PP5 becomes activated when the autoinhibitory mechanism is released by interaction of the TPR region with Ca$^{2+}$/S100 proteins[20] or heat-shock protein 90 (Hsp90)[21,22]. Activation is also possible through binding of polyunsaturated fatty acids (e.g., arachidonic acid) or long chain fatty acid-CoA esters (LCACE)[23]. Mice deficient in PP5[24] and those overexpressing PP5 in the heart[25] are viable and grow normally, but do show some (more subtle) functional changes in cell types such as fibroblasts[24] or cardiomyocytes[25]. However, little else is known about PP5 in the heart and nothing in human cardiomyocytes.

An important function of PP5 is that of a negative feedback regulator of MAPKs. PP5 inhibits the stress-activated protein kinase JNK/p38 pathway by binding and dephosphorylating

apoptosis signal-regulating kinase-1 (ASK1, encoded by *MAP3K5*) in response to oxidative stress[26]. Moreover, PP5 dephosphorylates Raf-1 and suppresses downstream signaling to MAPK/ERK kinase (MEK, encoded by *MAP2K*) and ERK1 (*MAPK3*)/ERK2 (*MAPK1*)[27]. While PP5 also interacts with ERK1/2, it does not dephosphorylate it[28]. The relationship between PP5 and the MAPK/ERK branch is particularly interesting in connection with titin, since Raf-1, MEK1 (*MAP2K1*)/MEK2 (*MAP2K2*) and ERK2 are recruited to N2Bus via a linker, four-and-a-half-LIM-domains protein-1 (FHL-1)[29]. The N2Bus-FHL-1-MAPK complex forms a putative sarcomeric mechanosensor involved in pro-hypertrophic cardiac signaling[29]. Whereas in vitro evidence suggested that FHL-1 blocks phosphorylation of N2Bus by ERK2[14], phosphorylation of cardiac fibers by ERK2 readily lowered their titin-based passive tension[30].

Here we show that PP5 dephosphorylates titin specifically at N2Bus and reverses the effect of phosphorylation on titin-based passive tension. Through association with Hsp90 and components of the N2Bus-mechanosensor complex, PP5 operates in a compartmentalized manner at the sarcomeric I-bands. Our work establishes PP5 as a well-controlled enzyme in cardiomyocytes, which regulates titin properties and MAPK signaling to support dynamic heart function.

## Results

**PP5 binds the titin N2Bus domain**. In a yeast-2-hybrid screen (Y2H), we used full-length human titin N2Bus as bait and a human adult heart cDNA library as prey. Approximately 150 clones grew on selective media plates and 50 of them were tested positive in a β-galactosidase assay. Among the detected molecules were FHL-2, which was previously identified as a binding partner of N2Bus[31], and the catalytic domain of PP5 (PP5c), specifically, amino acids 205–266 of human PP5. The interaction with PP5 was verified in a direct Y2H binding assay using N2Bus as bait and PP5c or full-length PP5 as prey (Fig. 1a). Analysis of diploid clones by PCR revealed the expected inserts of 1.7 kb for N2Bus and 1.5 or 1.0 kb for PP5 or PP5c, respectively. The N2Bus-PP5 interaction was also confirmed in vitro by GST-pulldown assays, in which both full-length PP5 and PP5c interacted with N2Bus (Fig. 1b). Two other I-band titin regions located near N2Bus, PEVK and N2A, did not interact with PP5. To narrow down the interaction site for N2Bus on PP5, we used two PP5 deletion constructs in the GST-pulldown assays, the TPR region at the PP5 N-terminus ('T'-construct, residues 28–129) and the TPR region with additional amino acids reaching into the catalytic domain ('T$^+$'-construct, residues 28 to 211) (Fig. 1a). The T-construct showed no interaction with N2Bus, whereas T$^+$ did (Fig. 1b), suggesting that the site of interaction involves the N-terminal part of the catalytic domain. Taken together with the Y2H binding data, it appears that amino acids 205–211 of PP5 are critical for the interaction with N2Bus. These residues belong to an α-helix exposed at the outer region of PP5 and are therefore readily accessible[19]. The PP5-N2Bus binding was also probed by co-immunoprecipitation in HEK cells. PP5-HA was coupled to anti-HA agarose beads, N2Bus-myc-containing whole-cell lysate was added and bound protein eluted for detection by western blot. A weak but consistently reproducible interaction was observed (Fig. 1c), confirming that PP5 is a novel interactor of titin at the N2Bus.

**PP5-binding to N2Bus is enhanced by phosphorylation of N2Bus**. To test if PP5c binds to titin in muscle sarcomeres, we first used an assay on single isolated human cardiomyofibrils placed under an inverted microscope and stretched under the control of a piezo motor. Recombinant PP5c was added to the

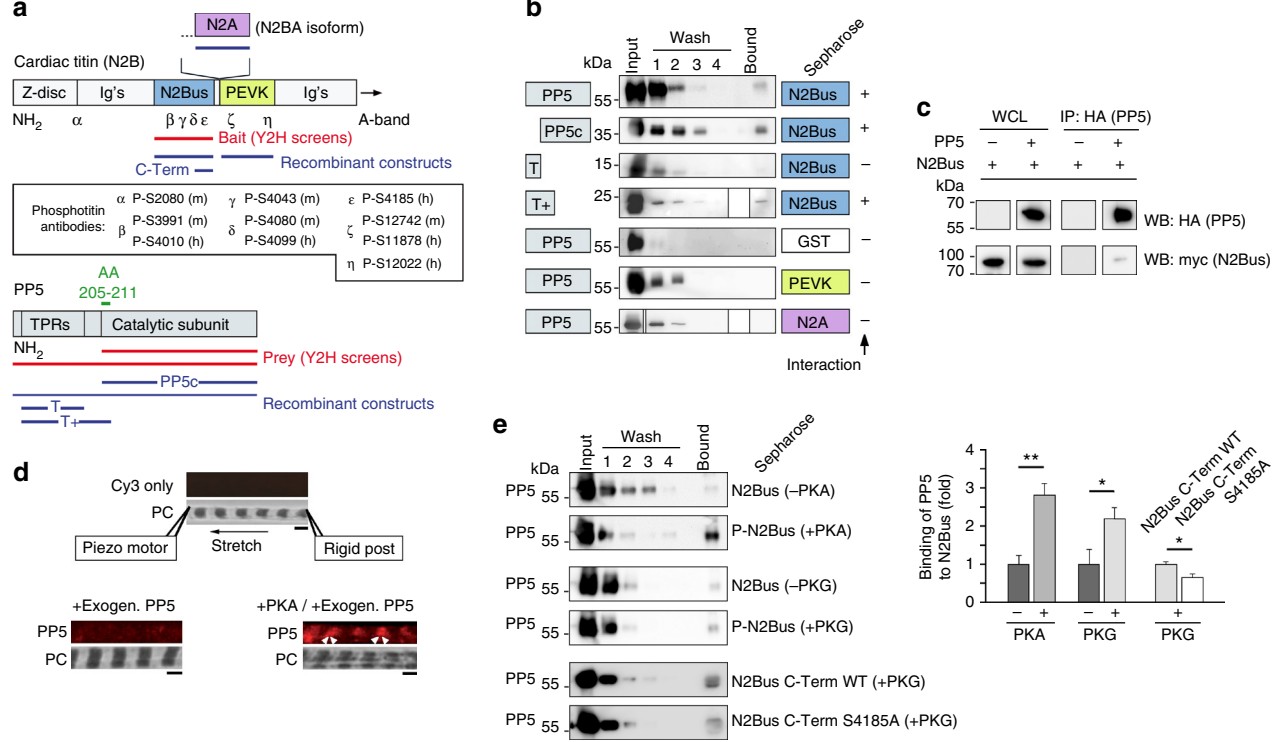

**Fig. 1** The elastic N2Bus domain of titin interacts with PP5. **a** Domain arrangement in the Z-disk/I-band region of cardiac titin N2B/N2BA isoforms and in PP5. Constructs generated for yeast-two-hybrid (Y2H) screens marked in red, those for GST-pulldown assays in blue, and N2Bus-binding amino acids (AA) of PP5 in green. Ig's, immunoglobulin-like domains. Inset: Epitope positions of all phospho-titin antibodies used in this study. (m), anti-mouse; (h), anti-human. **b** Summary of results of GST-pulldown assays probing interaction of N2Bus with full-length PP5, PP5 catalytic subunit (PP5c), or N-terminal PP5 fragments (T; T+). PP5-binding to PEVK or N2A titin domains was also tested. GST, glutathione S transferase (for negative control). Each test was performed a minimum of two times, mostly three times, with identical results. **c** Demonstration of PP5-N2Bus association by co-immunoprecipitation assay. PP5 (HA-tag) immunoprecipitations (IP) and whole-cell lysates (WCL) from HEK cells analyzed by western blot for N2Bus (myc-tag) and PP5. PP5 -/+indicates absence/presence of PP5 in the assay. This test was performed three times, with identical results. **d** Binding of PP5 to sarcomeric I-bands is enhanced by phosphorylation. Top: the experimental design for the stretching of single myofibrils and immunofluorescence image of stretched human cardiac myofibril incubated in relaxing buffer with Cy3-conjugated secondary antibodies alone (control), as well as phase-contrast image (PC). Bottom: representative images of myofibrils incubated with exogenous PP5c and stained against PP5c. The myofibril on the right was incubated with catalytic subunit of PKA before PP5c-treatment (arrowheads, I-band localization of PP5). Binding visualized by anti-PP5c primary and Cy3-conjugated secondary antibodies. Similar results were obtained from four other myofibrils per group. Bars, 2 µm. **e** Results of GST-pulldown assays probing interaction of PP5 with unphosphorylated N2Bus or N2Bus phosphorylated by PKA/PKG, as well as wildtype (WT) and S4185A mutant of C-terminal N2Bus fragment, both phosphorylated by cGMP-activated PKG. Left: representative immunoblots using anti-PP5 antibody. Right: relative signal intensities in 'Bound' lane, normalized to the mean intensity of the respective non-phosphorylated/C-Term WT control. Data are mean ± s.e.m., n = 4 assays/condition. *p < 0.05 and **p < 0.01, by two-tailed Student's t-test

myofibril and binding detected by indirect immunofluorescence. Consistent with the results of the co-immunoprecipitation assay (Fig. 1c), exogenous PP5c bound only weakly to the myofibrils (Fig. 1d, lower left images). However, the binding was much intensified and clearly localized to the sarcomeric I-bands when the myofibril was phosphorylated (by catalytic subunit of PKA) prior to incubation with PP5c (Fig. 1d, lower right images). In control experiments, neither Cy3-conjugated secondary antibody alone (Fig. 1d, top) nor recombinant green fluorescence protein[32] interacted with the myofibrils, suggesting that PP5 binding was not due to a generally increased stickiness of stretched sarcomeres. Considering the effect of prior phosphorylation of N2Bus on PP5 binding, we performed GST-pulldown assays with recombinant human PP5 and N2Bus, in which the latter was phosphorylated by PKA catalytic subunit or cGMP-activated PKG, or was left unphosphorylated (Fig. 1e). PP5 bound 2–3 times stronger to N2Bus if the titin region was phosphorylated (Fig. 1e, bar graph). Furthermore, in a 162-residue C-terminal fragment of N2Bus we mutated a known PKG-dependent phosphoserine at position 4185 (reference to full-length human

titin[13]) to alanine and used the wildtype (WT) and S4185A N2Bus constructs, phosphorylated by cGMP-activated PKG, in GST-pulldown assays with PP5. A small but significant decrease in binding strength was found for the S4185A mutant (Fig. 1e), providing further evidence that the PP5-N2Bus interaction is enhanced by phosphorylation of N2Bus.

**PP5 translocates to sarcomeres if induced in cardiomyocytes.**
In testing for PP5 expression in vivo, we found a relatively high level in embryonic (E18) and newborn (P1) rat hearts, which decreased to half in adult rat hearts (Fig. 2a). We thus prepared primary cultures of neonatal rat ventricular myocytes (NRVM) and studied the intracellular localization of PP5 before and after pharmacological manipulation using PP5-activator arachidonic acid (aa; 200 µM) or potent inhibitor okadaic acid (oa; 10 nM)[33]. Interestingly, PP5 activation by aa also increased the expression level of PP5, compared to untreated NRVM, whereas PP5 inhibition by oa decreased it (Fig. 2b). Confocal microscopy showed PP5 to be distributed in a diffuse pattern all over the cytoplasm in

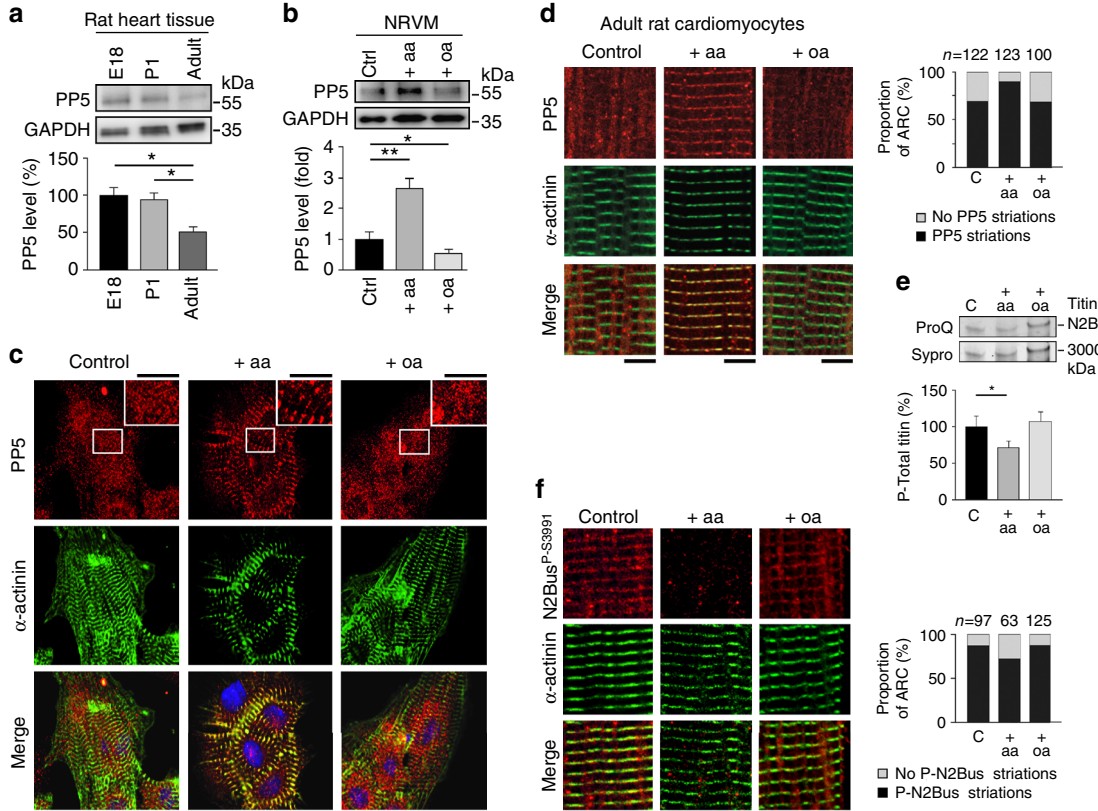

**Fig. 2** Endogenous PP5 and phospho-titin in rat cardiomyocytes. **a** PP5 expression by western blot in fetal (E18), newborn (P1) and adult rat heart tissue. PP5 indexed to GAPDH. Data are mean ± s.e.m., $n = 5$; *$p < 0.05$, by two-tailed Student's $t$-test. **b** PP5 expression by western blot in neonatal rat ventricular myocyte (NRVM) cultures under baseline conditions (control) and following treatment with arachidonic acid (aa; 200 μM, 2 h) or okadaic acid (oa, 10 nM, 1 h). PP5 indexed to GAPDH. Data are mean ± s.e.m., $n = 5$ (three different cell culture batches); *$p < 0.05$ and **$p < 0.01$, by Bonferroni adjusted $t$-test. **c** PP5 localization in control, aa-treated, and oa-treated NRVM cultures by indirect immunofluorescence. PP5 antibody (secondary antibody: Cy3-conjugated IgG), counterstained with α-actinin antibody (secondary antibody: FITC-conjugated IgG). The merged image also shows staining of nuclei using Hoechst. Bars, 10 μm. **d** PP5 localization in control, aa-treated, and oa-treated adult rat cardiomyocyte (ARC) cultures by indirect immunofluorescence. The same antibodies as in **c** were used. Bars, 5 μm. Right bar graph shows proportion of ARC exhibiting clear PP5 striations or no such striated pattern, for each group. Numbers above columns indicate total number of cells included in the analysis. **e** Total titin phosphorylation in the three ARC groups measured by ProQ Diamond phosphoprotein vs. Sypro Ruby total protein stain. Bar graph shows mean ± s.e.m., $n = 9$ (from three independent cell preparations); *$p < 0.05$, by two-tailed Student's $t$-test. **f** Localization of phosphoserine P-S3991 (titin N2Bus) in control, aa-treated, and oa-treated ARC cultures by indirect immunofluorescence, using anti-N2Bus[P-S3991] antibody (secondary antibody: Cy3-conjugated IgG), counterstained with α-actinin antibody (secondary antibody: FITC-conjugated IgG). Bars, 5 μm. Right bar graph shows proportion of ARC exhibiting clear N2Bus[P-S3991] striations or no such striated pattern, for each group. Numbers above columns indicate total number of cells included in the analysis

control NRVM, with only an occasional hint at a more regular striation pattern (Fig. 2c). However, stimulation by aa typically caused some PP5 to translocate to the sarcomeric Z-disk/I-band region, as suggested by co-localization with α-actinin (Fig. 2c; note that due to the short sarcomere length (SL) of NRVM cultures, the I-band and Z-disk regions are hardly distinguishable by confocal microscopy). Treatment with oa always resulted in PP5 staining patterns with no signs of regular striations. Because PP5 negatively regulates MAPK signaling by dephosphorylating Raf1[27], we also measured the expression and activity (phosphorylation) of the MAPK effector kinases ERK1/2 in NRVM. Whereas ERK1/2 expression was unaltered by aa and oa treatment, phospho-ERK1/2 was reduced by 65% in aa-treated cells but little affected by oa, as compared to controls (Supplementary Fig. 1a). Inhibitor of Raf1 (Raf1-I; 20 μM) lowered phospho-ERK1/2 by ~ 50%. Even larger alterations in ERK1/2 activity were seen in NRVM when the MAPK pathway was first stimulated by angiotensin-2 (AngII) or endothelin-1 (ET-1), prior to treatment with aa, oa, or Raf1-I (Supplementary Fig. 1b). Following ERK1/2 activation by AngII, treatment with oa now had an additional stimulatory effect on phospho-ERK1/2. Taken together, these

findings suggest a link between PP5 and MAPK signaling, for the first time also in cardiomyocytes.

We also studied PP5 localization in primary cultures of adult rat cardiomyocytes (ARC) (Fig. 2d). Before PP5 induction, the phosphatase was diffusely cytosolic in some cells, but appeared in a regular striated (sarcomeric) pattern in others. Quantitation of this distribution showed that the proportion of cells with regular PP5 striations increased substantially in aa-treated (PP5-stimulated) compared with control ARC (Fig. 2d, bar graph). In most aa-stimulated cells, PP5 co-localized with the Z/I-region of the sarcomere marked by α-actinin. Treatment with oa did not alter the proportion of cells with regular PP5 striations, in comparison to controls, and the diffuse PP5 distribution was seen more frequently than in aa-treated cells (Fig. 2d). We conclude that, particularly after induction in cardiomyocytes, PP5 is preferentially translocated to the sarcomeres.

**PP5 dephosphorylates titin at the N2Bus element.** In order to determine whether PP5 dephosphorylates titin, we first quantified the total titin phosphorylation level in ARC cultures. Induction of

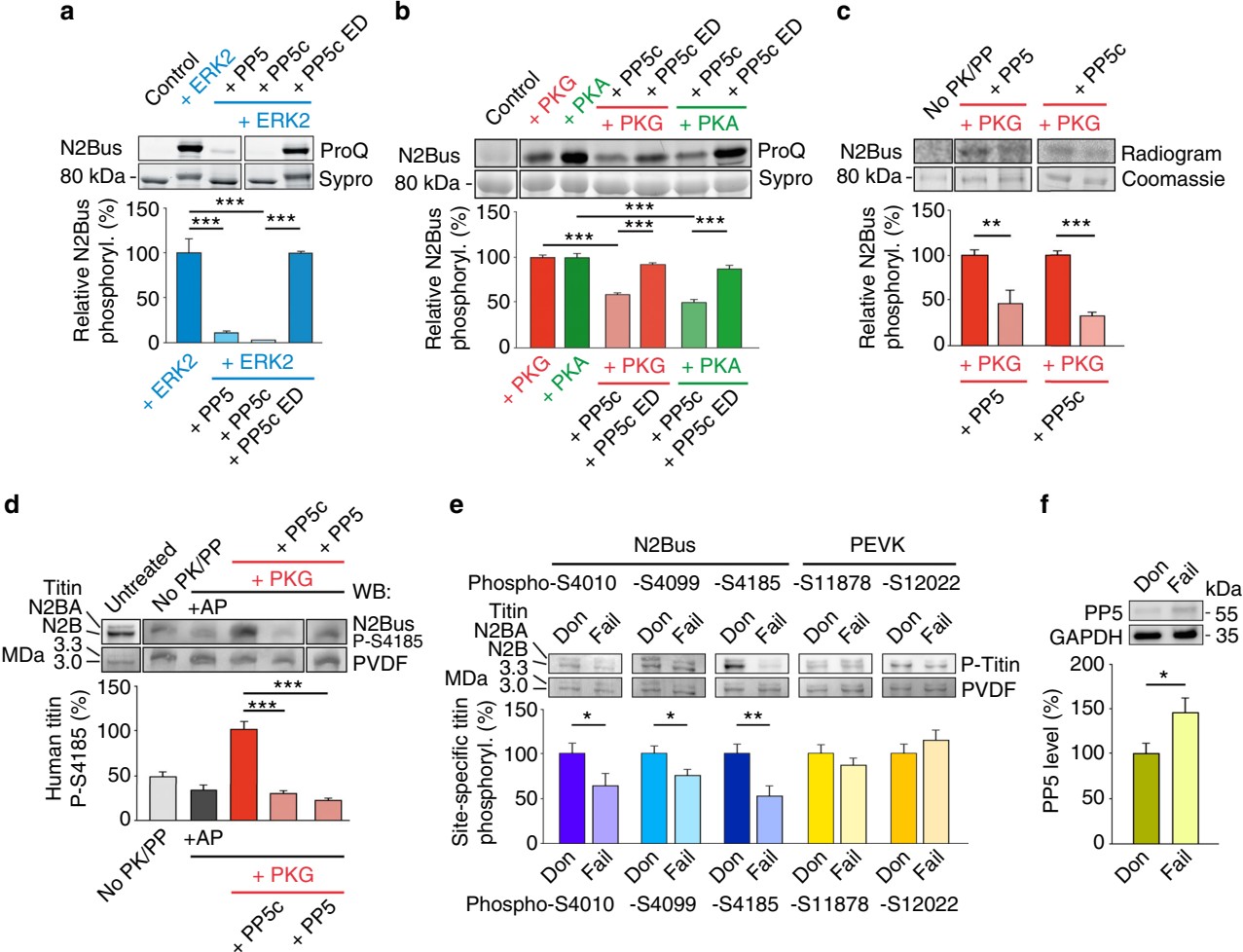

**Fig. 3** PP5 dephosphorylates recombinant and endogenous human N2Bus. **a, b** ProQ Diamond phosphoprotein vs. Sypro Ruby total protein stain of human recombinant N2Bus phosphorylated (in presence of ATP) by ERK2 (**a**), catalytic subunit of PKA (**b**), or cGMP-activated PKG (**b**), and effect of human recombinant full-length PP5, PP5 catalytic subunit (PP5c), or enzymatic dead (ED) PP5c mutant H304A on phosphorylation. 'Control' is recombinant construct but no ATP/enzyme. $n = 5$. **c** $^{32}$P-ATP autoradiography using N2Bus phosphorylated by cGMP-activated PKG and dephosphorylated by PP5 or PP5c. 'No PK/PP' is control in presence of $^{32}$P-ATP, but no enzyme. $n = 4$. **d** Western blot (WB) of permeabilized human (donor) heart tissue treated with various enzymes, as follows: alkaline phosphatase (AP), cGMP-activated PKG, then PP5 or PP5c. Detection of titin phosphorylation with phospho-specific antibody to P-S4185 in human N2Bus. PVDF was coomassie-stained to reveal loading on gel. WB signals were normalized to PVDF signals. $n = 4$. **e** Site-specific phosphorylation of residues within N2Bus and PEVK titin regions by western blot in myocardial tissue from human donor (Don) and end-stage failing (Fail) hearts ($n = 10$/group). Detection with phospho-specific antibodies to P-S4010, P-S4099, P-S4185, P-S11878, and P-S12022 in the human titin sequence. WB signals were normalized to PVDF signals. **f** Mean PP5 expression in human donor vs. end-stage failing hearts ($n = 10$/group). In **a–f**, bar graphs show relative phosphorylation indexed to the respective phosphorylated controls. Data are mean ± s.e.m.; *$p < 0.05$, **$p < 0.01$, and ***$p < 0.001$, by two-tailed Student's $t$-test or Holm-Sidak method

PP5 by aa reduced titin phosphorylation by a small but significant amount, whereas oa treatment did not alter it, in comparison to non-treated control cells (Fig. 2e). Next we performed indirect immunofluorescence on ARC using a phospho-specific antibody against rat/mouse phosphoserine P-S3991 located in titin's N2Bus element (equivalent to P-S4010 in human titin[15]), which is known to be phosphorylated by ERK2[14] and PKA[34]. In most control cells, the N2Bus^P-S3991 epitope appeared in a regular striation pattern, co-localizing with the Z/I-region of the sarcomere marked by α-actinin (Fig. 2f). In PP5-stimulated (aa-treated) ARC, the proportion of cells with regular N2Bus^P-S3991 striations was reduced compared to controls, whereas oa treatment did not alter this proportion (Fig. 2f, bar graph). These results suggest that induction and translocation of PP5 to the cardiac sarcomeres may result in dephosphorylation of titin at N2Bus.

To substantiate this finding, we generated recombinant human N2Bus and tested whether recombinant PP5 dephosphorylates the construct after phosphorylation by a kinase. ERK2-phosphorylated N2Bus was dephosphorylated by full-length PP5 or (most effectively) by PP5c, which induced >90% reduction of phosphorylation (Fig. 3a). In contrast, enzymatic dead catalytic subunit of PP5 (PP5c^H304A ED) was unable to dephosphorylate the ERK2-phosphorylated N2Bus. Similar experiments were conducted using PKA catalytic subunit or cGMP-activated PKG as the phosphorylating kinases. Under these conditions, PP5c dephosphorylated N2Bus by 45–50%, whereas PP5c ED again showed no dephosphorylation effect (Fig. 3b). These results, obtained using ProQ Diamond/Sypro Ruby staining, were further validated in $^{32}$P-autoradiography experiments, which showed that full-length PP5 and PP5c dephosphorylated the PKG-phosphorylated N2Bus by ~55%

and ~65%, respectively (Fig. 3c). The relatively high phosphatase activity of full-length PP5 found in these experiments was unexpected but consistently observed; it might be due to partial cleavage (and thus, activation) of the phosphatase in vitro or incomplete autoinhibition.

Recombinant full-length PP5 or PP5c also dephosphorylated endogenous N2Bus in skinned (demembranated) human heart tissue (Fig. 3d). Here, titin phosphorylation was detected by western blot using phosphosite-specific antibodies to P-S4185, a PKG/PKA-phosphorylated serine in human N2Bus[13]. Alkaline phosphatase (AP) was used to initially dephosphorylate the heart samples, before exogenous cGMP-activated PKG was given to maximize N2Bus phosphorylation (=100% phosphorylation). Subsequently, full-length PP5 or PP5c were able to dephosphorylate N2Bus$^{\text{P-S4185}}$ by 70% to 80% (Fig. 3d). These results demonstrate that PP5 dephosphorylates N2Bus in vitro and in heart tissue.

**PP5 is increased in human and experimental heart failure**. In keeping with our earlier findings[34,35], western blotting using phospho-specific antibodies to human titin (for epitope positions, see Fig. 1a) at residues P-S4010, P-S4099 (a PKG-dependent phosphosite)[34], and P-S4185 detected a phosphorylation deficit at all three of these N2Bus sites in human end-stage failing hearts vs. non-failing donor hearts, which amounted to ~25–50% (Fig. 3e). In contrast, phosphorylation at P-11878 and P-12022 within the titin PEVK element was unaltered in failing vs. donor hearts. Interestingly, PP5 expression was ~45% higher in the failing hearts (Fig. 3f). Furthermore, in hearts from elderly hypertensive dogs with diastolic dysfunction, we observed ~30% less phosphorylation at N2Bus site P-S4010, compared to healthy canine hearts (Supplementary Fig. 2a), whereas PP5 expression was ~60% higher in the diseased hearts (Supplementary Fig. 2b). We conclude that hypo-phosphorylation of N2Bus in failing myocardium from humans and animal models, which is known to elevate titin-based passive stiffness[3], may be attributable, at least in part, to increased PP5-mediated dephosphorylation.

**PP5 transgenic mouse hearts are hypo-phosphorylated at N2Bus**. The increase in PP5 expression observed in failing human hearts prompted us to study titin phosphorylation in a transgenic (TG) mouse model with cardiac-specific overexpression of PP5. The cardiac phenotype of the TG model is mild, but some impairment of cardiac contractility has been noted[25]. The PP5 protein level of the TG hearts was increased, on average by a factor of 7–8, compared to that of matched WT hearts (Fig. 4a). As expected due to the fact that PP5 dephosphorylates Raf1 at S338[27], phospho-Raf1$^{\text{S338}}$ was significantly reduced in PP5 TG hearts (Fig. 4a). Immunostaining against endogenous PP5 on myocardial tissue sections demonstrated that the phosphatase was present at low levels in WT cardiomyocytes and was distributed in a diffuse cytoplasmic pattern, with an occasional sarcomeric I-band localization (Fig. 4b, c). In PP5 TG myocytes, the phosphatase localized in a regular striated pattern to the I-bands near the PEVK domain of titin and thus, the N2Bus position, as shown by immunofluorescence (Fig. 4c) and immunoelectron microscopy (Fig. 4b). Quantitation of the distribution of nanogold particles on immunoelectron micrographs indicated the presence of ~3 times more particles in PP5 TG than in WT hearts and suggested that ~65% of total PP5 protein localized to the I-band springs in PP5 TG cardiomyocytes, compared to ~30% in WT cells (Fig. 4b, bar graph).

Total titin phosphorylation was measured by ProQ Diamond/Sypro Ruby staining and a trend for reduced phosphorylation was observed in PP5 TG vs. WT mouse hearts (Fig. 4d, upper left).

Site-specific phosphorylation was detected by western blot using a panel of phosphoserine-specific antibodies against mouse titin (for epitope positions, see Fig. 1a). Each phospho-specific antibody was used together with a corresponding sequence-specific antibody (Pan), which helped ascertain equal protein loading. All three phosphoserines studied in N2Bus, P-S3991 (ERK2/PKA-dependent), P-S4043 (CaMKIIδ-dependent) and P-S4080 (PKG-dependent), were significantly hypo-phosphorylated (reduction, 45–50%) in PP5 TG vs. WT hearts (Fig. 4d, right panels). In contrast, a phosphoserine at the Z-disk/I-band junction of titin (P-S2080) showed unaltered phosphorylation, as did P-S12742 in the PEVK domain (which is P-S11878 in humans). We also used the anti-P-S3991 antibody for immunostaining on tissue sections and found regularly spaced doublet lines for N2Bus$^{\text{P-S3991}}$ signals in WT co-localizing with PEVK, but low-intensity N2Bus$^{\text{P-S3991}}$ signals in PP5 TG (Fig. 4e). On immunoelectron micrographs, most nanogold particles indicative of N2Bus$^{\text{P-S3991}}$ were found at the I-band springs in WT, whereas only a small number was at the I-bands in PP5 TG hearts (Fig. 4f). These data led us to conclude that, in vivo, PP5 specifically dephosphorylates phosphoserines of titin located in the N2Bus region.

**PP5 transgenic cardiomyocytes have increased passive tension**. Passive force ($F_{\text{passive}}$) measurements were performed on skinned single cardiomyocytes of WT and PP5 TG mouse hearts in relaxing buffer. Cell stretch over the range 1.8–2.4 μm SL resulted in the typical, quasi-exponential increase in $F_{\text{passive}}$ (Fig. 5a). Importantly, the $F_{\text{passive}}$–SL curves were much steeper in PP5 TG than in WT cardiomyocytes, and mean $F_{\text{passive}}$ was significantly increased at SL ≥ 2.0 μm (Fig. 5b, c). When PP5 TG and WT cardiomyocytes were incubated with recombinant PP5c, WT cells showed an additional increase in $F_{\text{passive}}$, whereas TG cells were not altered in their stiffness (Fig. 5b). Moreover, $F_{\text{passive}}$ could be reduced significantly in both WT and PP5 TG cardiomyocytes upon treatment with ERK2 (Fig. 5c, d), PKA catalytic subunit (Supplementary Fig. 3a), or cGMP-activated PKG (Supplementary Fig. 3b, c). In the presence of exogenous kinase, $F_{\text{passive}}$ was always lower in WT than in TG cells. Finally, incubation of WT myocytes with recombinant PP5c following treatment with ERK2 or PKG (and washout of the kinase) significantly increased $F_{\text{passive}}$ to a level well above that measured in untreated WT cells (Fig. 5d and Supplementary Fig. 3c). We conclude that PP5 is an antagonist to the various PKs that phosphorylate N2Bus, in terms of its effect on cardiomyocyte $F_{\text{passive}}$.

**PP5 and N2Bus bind to components of an I-band mechanosensor**. N2Bus interacts directly with small heat-shock proteins[32] and FHL proteins, which on their part recruit metabolic enzymes (FHL-2[31]) and the MAPKs Raf1, MEK1/2 and ERK2 (FHL-1[29]) to N2Bus, thus forming a putative mechanosensor complex. We studied by GST-pulldown assay whether PP5 also interacts with FHL-1 and found binding with both full-length PP5 and PP5c (Fig. 6a, top). In cardiomyocytes, FHL-1 mainly localized to the sarcomeric I-bands (marked by PEVK antibody) and thus, at the expected N2Bus binding site (Supplementary Fig. 4a). There was no difference in FHL-1 localization between PP5 TG and WT hearts. Additional in vitro binding assays revealed that PP5 or PP5c also interacted with ERK2 and Hsp90 (Fig. 6a), both of which have previously been described as binding partners of PP5[28,36]. Since PP1 and PP2a are important phosphatases in cardiomyocytes, we determined their expression levels, as well as that of ERK2, in PP5 TG vs. WT mouse hearts, but observed no differences (Supplementary Fig. 4b).

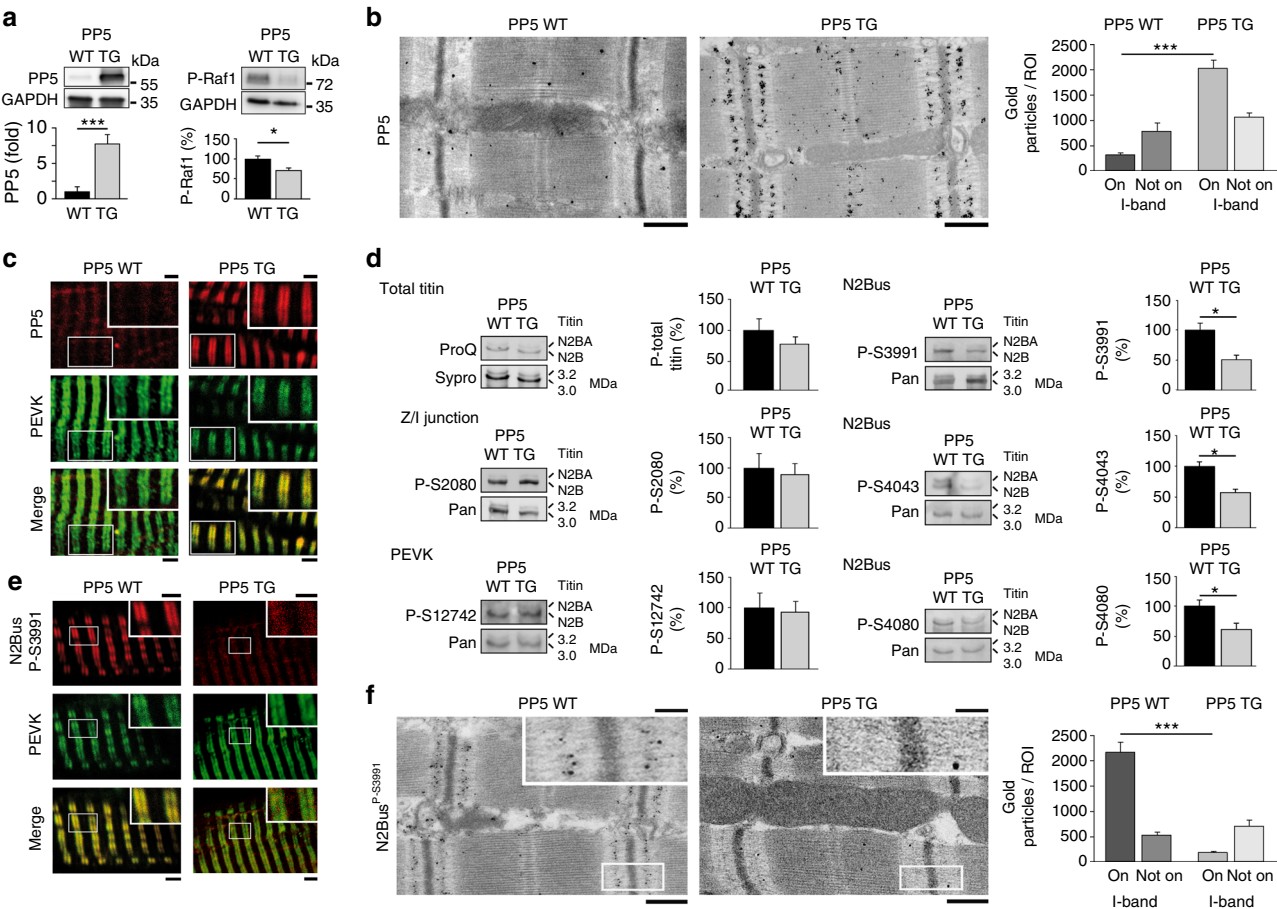

**Fig. 4** PP5 and phospho-titin in PP5-overexpressing TG mouse hearts. **a** Expression level of PP5 (left) and phospho-Raf1[S338] (right) in PP5 TG vs. WT mouse hearts by western blot. mean ± s.e.m., $n = 4$ hearts/group (age 5–6 months); duplicate analysis/group. **b** Sarcomeric localization of PP5 in PP5 WT and TG hearts by immunogold electron microscopy. Bars, 500 nm. Bar graph shows average number of gold particles counted in 50-μm$^2$-sized regions-of-interest (ROI), either on the sarcomeric I-band or elsewhere in the cardiomyocyte ('Not on I-band'). Data are mean ± s.e.m., $n = 5$ ROIs from 2 hearts/group. **c** PP5 localization in cardiomyocytes from PP5 TG and WT mouse hearts by indirect immunofluorescence. PP5 antibody (secondary antibody: Cy3-conjugated IgG), counterstained with anti-PEVK (titin) antibody (secondary antibody: FITC-conjugated IgG). Bars, 2 μm (main) and 1 μm (insets). **d** PP5 overexpression specifically decreases titin phosphorylation at N2Bus in PP5 TG vs. WT hearts. Total titin phosphorylation measured by ProQ Diamond/Sypro Ruby staining (upper left), site-specific titin phosphorylation detected by western blot using antibodies to P-S3991, P-S4043, P-S4080 (all N2Bus; right panels), P-S2080 (titin Z/I junction), and P-S12742 (PEVK region). Phospho-titin signals were normalized to total titin signals detected by WB using a panel of sequence-specific antibodies (Pan). Means were indexed to those of control (WT) groups. Data are mean ± s.e.m., $n = 4$ hearts/group, samples analyzed in triplicate. **e** Localization of phospho-N2Bus[P-S3991] in cardiomyocytes from PP5 TG and WT hearts by indirect immunofluorescence. Anti-N2Bus[P-S3991] antibody (secondary antibody: Cy3-conjugated IgG), counterstained with anti-PEVK antibody (secondary antibody: FITC-conjugated IgG). Bars, 2 μm (main) and 1 μm (insets). **f** Sarcomeric localization of phospho-N2Bus[P-S3991] in PP5 WT and TG hearts by immunogold electron microscopy. Bars, 500 nm (main) and 100 nm (insets). Bar graph shows average number of gold particles counted in 50-μm$^2$-sized regions-of-interest (ROI), either on the sarcomeric I-band or elsewhere in the cardiomyocyte ('Not on I-band'). Data are mean ± s.e.m., $n = 5$ ROIs from 2 hearts/group. In **a** and **d**, bar graphs show relative signal changes indexed to the respective controls. In **a**, **b**, **d**, and **f**, *$p < 0.05$ and ***$p < 0.001$, by two-tailed Student's $t$-test

Focusing on interactors of N2Bus, we confirmed binding of this titin region to FHL-1[29] and FHL-2[31] by GST-pulldown assay (Fig. 6b). FHL-2 also binds ERK2[37] and could thus play a role in the N2Bus-associated mechanosensor. Interestingly, N2Bus interacted in vitro with the α and β isoforms of Hsp90 (Fig. 6b). Since Hsp90 activates PP5 through binding to the TPR region of the phosphatase, we performed in vitro 'competition' assays to test the impact of Hsp90 on the N2Bus-PP5 association. In these assays, pre-incubation of N2Bus-bound sepharose beads with PP5, followed by incubation with Hsp90, resulted in relatively weak binding of PP5 to N2Bus, whereas pre-incubation of N2Bus-bound sepharose beads with Hsp90, followed by incubation with PP5, resulted in significantly stronger binding of PP5 to N2Bus (Fig. 6c). Thus, binding of

Hsp90 to N2Bus could promote subsequent PP5 binding to N2Bus. In this context, we immunostained against Hsp90 on myocardial tissue sections from PP5 TG and WT mice and found a diffuse cytosolic distribution of the chaperone in WT, with only an occasional hint at a more regular (sarcomeric) striation pattern (Fig. 6d). However, in PP5 TG hearts, Hsp90 consistently showed a regular striation pattern, in addition to the cytosolic localization, and co-localization with PEVK suggested binding to I-band titin, presumably N2Bus (Fig. 6d). Thus, Hsp90 and PP5 (Fig. 4c) may translocate to N2Bus in a coordinated manner. In conclusion, we confirmed known binary interactions within the N2Bus-associated mechanosensor and provided evidence for additional interactions involving PP5 and some of its binding partners.

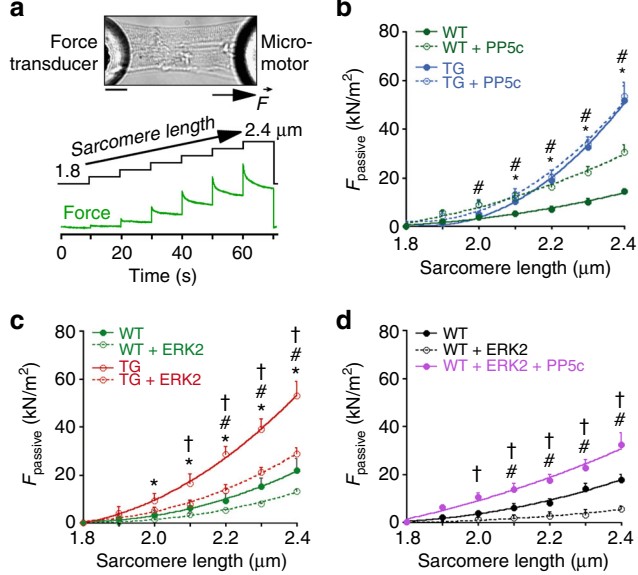

**Fig. 5** Passive tension of cardiomyocytes of WT and PP5 TG mouse hearts. **a** Representative image of permeabilized cardiomyocyte glued at the ends to a force transducer and micromotor, respectively, and experimental protocol. Bar, 20 μm. **b** Passive tension ($F_{passive}$) vs. SL curves of permeabilized single PP5 TG and WT cardiomyocytes in relaxing solution, before and after treatment with recombinant PP5c. **c** $F_{passive}$-SL curves of permeabilized single PP5 TG and WT cardiomyocytes before and after treatment with ERK2. **d** $F_{passive}$–SL curves of permeabilized single WT cardiomyocytes before and after treatment with ERK2 and additional exposure to PP5c. Data are mean ± s.e.m., $n = 5$ cells/condition (2 different hearts/condition). Curves are second-order polynomial fits to the means. *$p < 0.05$, TG vs. WT; #$p < 0.05$, WT + PP5c vs. WT in **b** and WT + ERK2 vs. WT in **c** and **d**; †$p < 0.05$, TG + ERK2 vs. TG in **c** and WT + ERK2 + PP5c vs. WT + ERK2 in **d**; all by two-tailed Student's $t$-test

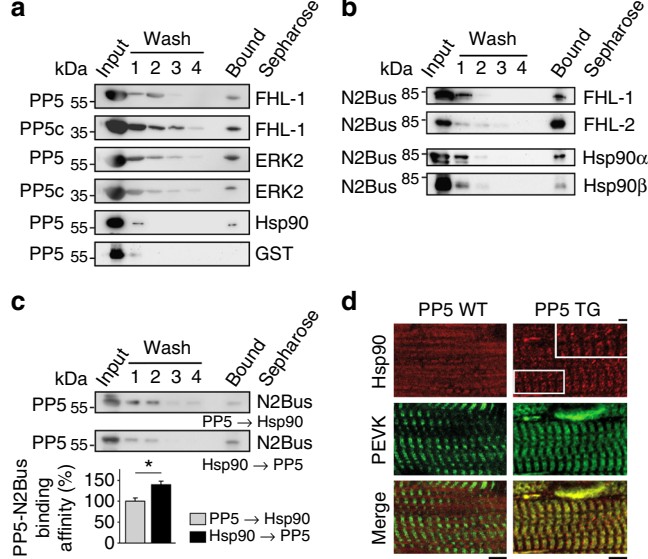

**Fig. 6** Binary interactions of PP5 or N2Bus and role of Hsp90. **a** Interactions of PP5 or PP5c by GST-pulldown assay. The PP5-GST assay is a negative control. **b** Interactions of N2Bus by GST-pulldown assay. **c** Impact of Hsp90 on PP5-N2Bus binding in GST-pulldown 'competition' assays. 'PP5→Hsp90': N2Bus immobilized on GSH-beads, incubated with PP5, then washed, and Hsp90 added thereafter. 'Hsp90→PP5': N2Bus immobilized on GSH-beads, incubated with Hsp90, then washed, and PP5 added thereafter. Summary data in bar graph are relative values indexed to the 'PP5→Hsp90' condition; mean ± s.e.m., $n = 5$ experiments/condition; *$p < 0.05$, by two-tailed Student's $t$-test. **d** Localization of Hsp90 in cardiomyocytes from PP5 TG and WT hearts by indirect immunofluorescence. Anti-Hsp90 antibody (secondary antibody: Cy3-conjugated IgG), counterstained with anti-PEVK (titin) antibody (secondary antibody: FITC-conjugated IgG). Bars, 5 μm (main) and 1 μm (inset)

**FHL-1-deficient hearts show reduced N2Bus phosphorylation.** In vitro data suggested that FHL-1 blocks ERK2-mediated phosphorylation of specific N2Bus residues, including S3991[14]. However, FHL-1 may also promote N2Bus phosphorylation, because FHL-1 is needed to anchor MAPKs at N2Bus[29]. To address this conundrum, we quantified cardiac titin phosphorylation in a mouse model deficient in FHL-1[29]. Total phospho-titin was significantly lowered by ~30%, whereas N2Bus phosphorylation at S3991 was reduced by ~50% in FHL-1 knockout (KO) vs. WT hearts (Fig. 7a). As a control, we measured phosphorylation of S2080 at the Z-/I-band junction of titin and found no difference between WT and FHL-1 KO (Fig. 7a). On immunofluorescently stained tissue sections of both WT and FHL-1 KO hearts, PP5 appeared mainly in the cytosolic space and sometimes at the sarcomeres; a difference in PP5-staining intensity and pattern was not consistently observed (Fig. 7b). Anti-P-S3991 antibody marked WT cardiomyocytes in a regular striated (doublet) pattern and at relatively high intensities, overlaying with PEVK, whereas FHL-1 KO cardiomyocytes consistently showed very low anti-N2Bus[P-S3991] staining intensity (Fig. 7c). In contrast, anti-P-S2080 antibody gave intense striation patterns in both WT and FHL-1 KO cardiomyocytes (Fig. 7d). On immunoelectron micrographs, N2Bus[P-S3991] nanogold particles localized abundantly to the sarcomeric I-band springs in WT but were nearly absent from FHL-1 KO cardiomyocytes (Fig. 7e), whereas anti-P-S2080 antibody again marked the Z/I titin region similarly in both WT and FHL-1 KO (Fig. 7f). These findings show that FHL-1 does not block phosphorylation of N2Bus in vivo (at least at ERK2/PKA-dependent P-S3991) and suggest

that FHL-1 supports phosphorylation specifically at N2Bus sites (but not I-band titin in general), presumably by targeting PKs to this spring element.

## Discussion

Unlike the phosphatases PP1, PP2a, and PP2b (calcineurin)[2,38], PP5 has gained little attention in the cardiac field, although it is readily expressed in cardiomyocytes. A reason may be the low basal activity of PP5 brought about by autoinhibition[18,19]. However, PP5 is activated when Hsp90, $Ca^{2+}$/S100 proteins, arachidonic acid or LCACE bind to the enzyme (Fig. 8), to release the autoinhibitory mechanism[20–23]. Our work now demonstrates an important role for this distinctly regulated phosphatase in cardiomyocytes, in that it binds and specifically dephosphorylates the N2Bus spring element in cardiac titin, thus acting antagonistic to the PKs that phosphorylate N2Bus and lower cardiomyocyte passive tension. PP5 performs this function as a novel constituent of the sarcomeric mechanosensor formed by N2Bus, FHL-1, and the MAPK family members Raf1, MEK1/2, and ERK2[29]. Unlike proposed earlier[14], FHL-1 does not block ERK2-mediated N2Bus phosphorylation in this complex structure, as demonstrated by us using WT and FHL-1 KO mouse hearts. Instead, FHL-1 supports efficient phosphorylation of specific serines in N2Bus, probably because it is important for the recruitment of MAPKs[29] (and perhaps other kinases) to N2Bus.

One effect of PP5 on the N2Bus-associated mechanosensor complex is brought about by dephosphorylation of Raf1, which blocks downstream MAPK signaling[27]. ERK2 cannot shuttle to

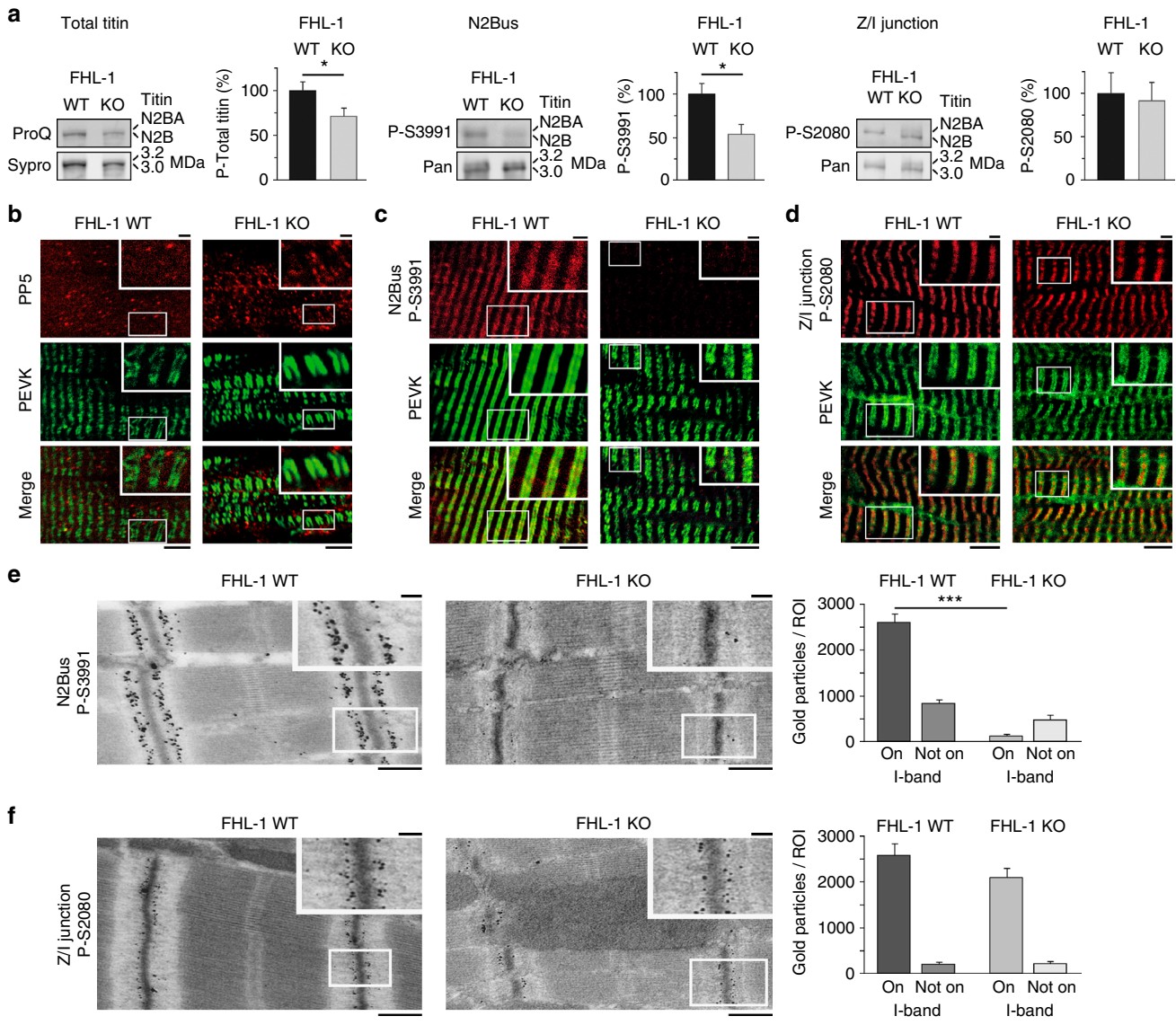

**Fig. 7** Reduced N2Bus phosphorylation in FHL-1-deficient cardiomyocytes. **a** Total titin phosphorylation measured by ProQ Diamond/Sypro Ruby staining (left) and site-specific titin phosphorylation detected by western blot using antibodies to P-S3991 (N2Bus; middle) or P-S2080 (Z/I junction; right). Site-specific titin phosphorylation levels were normalized to total titin levels detected by WB using sequence-specific antibodies (Pan). Means were indexed to those of control (WT) groups. Data are mean ± s.e.m., $n = 3$ hearts/group, samples analyzed in triplicate; *$p < 0.05$, by two-tailed Student's $t$-test. **b**, **c**, **d** Localization of PP5 (**b**), phospho-N2Bus$^{P-S3991}$ (**c**), and phospho-Z/I-junction$^{P-S2080}$ (**d**) in cardiomyocytes from FHL-1 WT and KO hearts by indirect immunofluorescence. Anti-PP5, anti-N2Bus$^{P-S3991}$ or anti-Z/I-junction$^{P-S2080}$ (secondary antibody: Cy3-conjugated IgG), counterstained with anti-PEVK antibody (secondary antibody: FITC-conjugated IgG). Bars, 5 μm (main) and 1 μm (insets). **e**, **f** Sarcomeric localization of phospho-N2Bus$^{P-S3991}$ (**e**) and phospho-Z/I-junction$^{P-S2080}$ (**f**) in FHL-1 WT and KO hearts by immunogold electron microscopy. Bars, 500 nm (main) and 100 nm (insets). Bar graph in **e** and **f** shows average number of gold particles counted in 50-μm²-sized regions-of-interest (ROI), either on the sarcomeric I-band or elsewhere in the cardiomyocyte ('Not on I-band'). Data in **e** and **f** are mean ± s.e.m., $n = 5$ ROIs from two hearts/group. In **a** and **e**, *$p < 0.05$ and ***$p < 0.001$, by two-tailed Student's $t$-test

the nucleus of the cardiomyocyte where it normally acts as a transcriptional co-factor[39] and mechanosensor function will be compromised (Fig. 8). In support of this scenario, Raf1 activity was reduced in the cardiomyocytes of PP5-overexpressing TG mice (Fig. 4a). Another effect of PP5 on the mechanosensor is triggered by the dephosphorylation of N2Bus. In PP5 TG hearts, we found lowered site-specific N2Bus phosphorylation and increased cardiomyocyte passive tension, compared to WT. Since the rise in passive tension follows from reduced distensibility of N2Bus due to the hypo-phosphorylation[13], and considering that mechanosensor function will depend on N2Bus distensibility[29], PP5 may inactivate the mechanosensor by stiffening N2Bus

(Fig. 8). Taken together, PP5-mediated regulation of mechanosensor activity is possible, in principle, by dephosphorylation of Raf1 or N2Bus, or both.

MAPK/ERK signaling via Gαq is promoted by G-protein coupled receptor (GPCR) agonists, such as ET-1 or AngII, in cardiomyocyte cultures and intact hearts (Fig. 8)[39–42]. Our results suggest that PP5 interferes with this pathway under physiological conditions in cardiomyocytes and regulates it in a compartmentalized manner. In cultured NRVM, we observed ERK1/2 activation by AngII or ET-1, hyper-activation of ERK1/2 in the presence of PP5-inhibitor okadaic acid, and very effective suppression of ERK activity by PP5-activator arachidonic acid

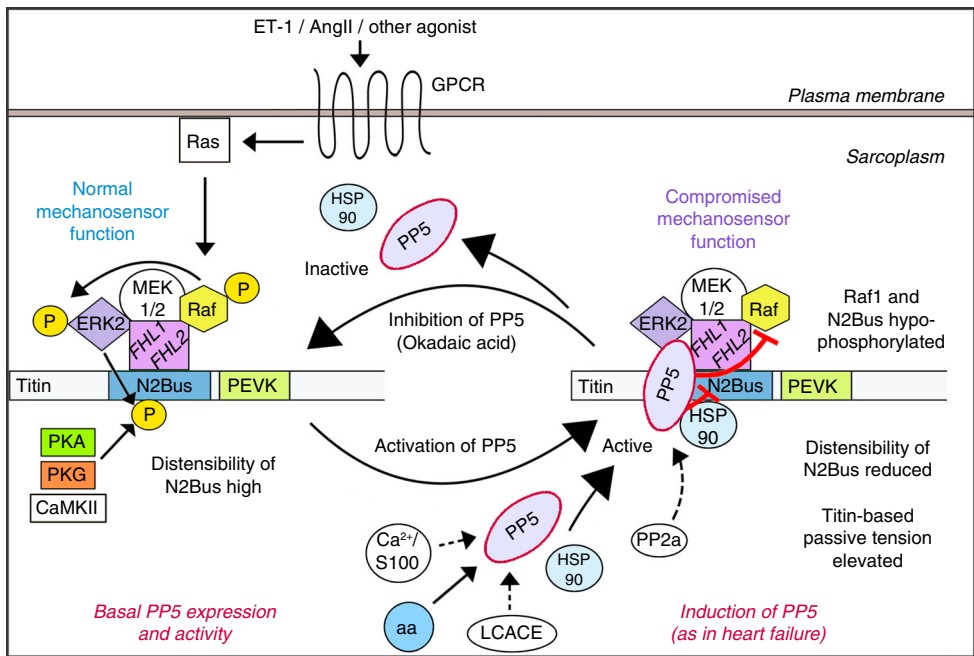

**Fig. 8** Proposed actions of PP5 in cardiomyocytes. Under basal conditions, PP5 activity towards titin N2Bus and MAPK/ERK family member Raf1 is low and the relatively high distensibility of N2Bus results in relatively low titin-based passive tension (left side). The strain-dependent mechanosensor connecting MAPKs to N2Bus via FHL-1 functions normally, as downstream signaling from Raf1 to ERK2 is enabled. When PP5 expression is increased (as in failing hearts) and PP5 becomes activated through interaction with Hsp90, $Ca^{2+}$/S100 protein, arachidonic acid (aa), or long chain fatty acid-CoA esters (LCACE), the phosphatase translocates to the I-band mechanosensor at N2Bus (right side). Thus, N2Bus (previously phosphorylated by ERK2, PKA, PKG, or CaMKII) is dephosphorylated, which reduces its distensibility and increases titin-based passive tension; the mechanosensor is now less sensitive. Raf-1 is also dephosphorylated and signaling to ERK2 is disabled, such that the mechanosensor function is additionally compromised. The process is embedded in signaling pathways activated via G-protein coupled receptor (GPCR) and Ras, and it can be reversed when PP5 is deactivated. (Molecules that have a color code were studied here, those with no color/white background were inferred from the literature)

(Supplementary Fig. 1). Cardiomyocytes treated with aa revealed enhanced sarcomeric I-band association of PP5 (Fig. 2), which could be the reason why the phosphatase appeared to have an increased half-life when activated (as indicated by elevated protein expression). Cardiomyocyte-restricted induction of PP5 in a TG mouse model also caused translocation of PP5 to the sarcomeres, specifically to the position of N2Bus (Fig. 4). Although the exact mechanism behind this translocation remains to be elucidated, our work suggested potential triggers. First, we found that phosphorylation of N2Bus enhanced PP5 binding to N2Bus in vitro, and the effect was partially suppressed by mutating a phosphoserine in N2Bus to alanine (Fig. 1d, e), suggesting the added negative charge at N2Bus was the reason for the increased PP5 affinity. Second, we found that PP5-activator Hsp90 translocated to the N2Bus region in PP5-overexpressing cardiomyocytes (Fig. 6d), just as did PP5. Because Hsp90 promoted PP5 binding to N2Bus in vitro (Fig. 6c) and itself interacted with N2Bus (Fig. 6b), the translocation of PP5 to N2Bus might be guided by Hsp90. In this context, it may be of relevance that the N2Bus binding site on PP5 is located not in the TPR region, but at the beginning of the catalytic subunit (Fig. 1a, b). Hsp90 can thus dock to the TPR region and activate PP5, while the phosphatase interacts with N2Bus. In summary, activated PP5 likely performs its physiological functions in cardiomyocytes, including the regulation of Gαq-mediated MAPK/ERK signaling, within a defined compartment at the sarcomeric I-bands (Fig. 8). Recruitment of PP5 to this compartment is promoted by stimulation via Hsp90 or aa (perhaps also $Ca^{2+}$/S100 and LCACE) and through phosphorylation of N2Bus.

Elevated PP5 expression was observed in end-stage failing human hearts and hypertensive dog hearts with diastolic

dysfunction (Fig. 3f and Supplementary Fig. 2b). Pathological triggers could thus activate and recruit PP5 to the sarcomeric N2Bus-FHL-MAPK complex in heart disease. In the failing human and dog hearts, we confirmed the reduced site-specific titin phosphorylation at N2Bus, but not PEVK (Fig. 3e and Supplementary Fig. 2a), reported earlier[34,35,43]. Such phosphorylation deficit at N2Bus is discussed as a main reason for the pathological increase in titin-based myocardial passive stiffness in HF, especially in HFpEF[3,13,44]. Whereas hypo-phosphorylation of N2Bus in HFpEF has so far been explained mainly in terms of pathologically downregulated cGMP-PKG signaling[3,45], our findings suggest another mechanism, which involves PP5. The phosphatase most efficiently dephosphorylated ERK2-phosphorylated human N2Bus in vitro, but also dephosphorylated PKA- or PKG-phosphorylated N2Bus (Fig. 3). Moreover, recombinant PP5c reversed the softening effect of ERK2 or cGMP-activated PKG on isolated skinned cardiomyocytes (Fig. 5d and Supplementary Fig. 3c). In PP5 TG mouse cardiomyocytes, ERK2/PKA-dependent N2Bus phosphosite P-S3991 was much less phosphorylated than in WT, as was CaMKIIδ-dependent P-S4043 and PKG-dependent P-S4080, but not PKCα-dependent P-S12742 in the PEVK domain or P-S2080 at the Z/I junction of titin (Fig. 4). Thus, the induction of PP5 in failing hearts and the activity of PP5 towards N2Bus (Fig. 3e) can explain why there is a phosphorylation deficit specifically at N2Bus but not at the PEVK domain[34,35,46,47].

These findings implicate several potential treatment strategies to reduce the high myocardial passive stiffness of HFpEF patients[44], which are worthwhile to be tested. First, the inhibition of PP5 could increase N2Bus phosphorylation and lower titin-based myocardial passive stiffness. Second, PP5 can associate with

PP2a, either directly[48] or indirectly through a regulatory subunit that binds both phosphatases[49]. PP2a is known to dephosphorylate multiple cardiomyocyte proteins[38,50] and it has been used experimentally to dephosphorylate titin in vitro[13]. Thus, one can speculate that PP5 may recruit PP2a to the sarcomeric I-band complex at N2Bus (Fig. 8), where these phosphatases could work in concert to dephosphorylate myofilament proteins like titin. If so, it will be interesting to test whether inhibition of PP2a benefits titin properties in normal and failing hearts. Moreover, PP5 could be linked structurally or functionally to other important phosphatases in cardiomyocytes, such as PP1 and calcineurin[2,38,51,52], which is a possibility worth studying in the future.

Finally, PP5 is strongly inhibited by spermine[17], a natural polyamine closely related to spermidine, which was recently shown to protect aging rodent hearts from diastolic dysfunction by softening the cardiac walls[53]. Importantly, the spermidine effect appeared to be related, in part, to increased titin phosphorylation at N2Bus. If spermine was found to have the same beneficial effects on the heart, an interesting potential therapeutic option for HFpEF patients would be to increase N2Bus phosphorylation and reduce titin-based stiffness by oral supplementation with this polyamine. Conversely, one can test spermidine for possible inhibitory effects on PP5. These novel approaches targeting PP5 should complement other efforts in HFpEF research directed at promoting the PKs (such as PKG[54]) that phosphorylate N2Bus and reduce cardiomyocyte passive stiffness.

In summary, we demonstrated an important signaling role for PP5 in cardiomyocytes, as the phosphatase binds and dephosphorylates the N2Bus region of titin, which affects cardiomyocyte passive stiffness. PP5 also interacts with components of the N2Bus-associated mechanosensor and blocks MAPK/ERK signaling in this complex. This way, PP5 functions in confined compartments at the sarcomeric I-bands. Reversing the induction of PP5 in heart failure would promote titin phosphorylation and help reduce pathologically increased diastolic stiffness.

## Methods

**Yeast-two-hybrid screens**. To search for interactors of N2Bus we first used an unbiased yeast-two hybrid binding assay, as described[55]. Human N2Bus titin region serving as bait was cloned into the pGBKT7 vector and transferred into yeast strain AH109 (Clontech). Human cardiac cDNA library (Clontech) serving as prey resided in a pACT2 vector, which was transformed into yeast strain Y187 (Clontech) using electroporation. Mating and screening procedures were carried out as described by the manufacturer (Clontech); 3-amino-1,2,4-triazol (70 mM) was used to eliminate background growth on selective media. Diploid yeast clones were let to grow on triple dropout plates. To exclude false-positive interactions, colonies were further analyzed for active β-galactosidase using an X-Gal filter-lift assay. Plasmids from blue-turned, positive clones were isolated and sequenced. A BLAST analysis revealed possible binding partners of N2Bus. To verify interactions, direct binding assays were performed in yeast with the potential binding partner (in pACT2 and Y187) and N2Bus (in pGBKT7 and AH109). Resulting diploid yeast clones from the small scale mating were analyzed by PCR for the presence of both fragments.

**Recombinant protein expression and GST-pulldown assays**. Human recombinant proteins were expressed as described[32]. Glutathione S-transferase (GST) fusion proteins were constructed by fusing a protein of interest to GST in the expression vector pGEX-4T-2 (Amersham) via EcoRI and XhoI. The expression of GST-fusion proteins was induced in Escherichia coli strain XL1blue (Stratagene) using 0.2 mmol/l isopropyl-β-D-thiogalactopyranoside (IPTG). The primers used for protein expression are listed in Supplementary Table 1. Only FHL-1 (Abnova) and ERK2 proteins (Abcam) were obtained from commercial sources. Purification was carried out according to manufacturer's instructions using glutathione sepharose beads (GSH, Amersham). Free polypeptides were obtained after thrombin cleavage (10 U) of GST-fusion proteins and spare thrombin was removed by using para-aminobenzamidine sepharose beads (Sigma-Aldrich). Purified proteins were used in multiple assays. Site-directed mutagenesis was used to generate mutant constructs of N2Bus (C-terminus) and PP5c. In the wildtype C-terminal N2Bus fragment (amino acids 411–572 of N2Bus) serine S4185 (referring to full-length human titin; UniProtKB entry Q8WZ42), which is phosphorylated by

PKG and PKA[13], was exchanged for an alanine (S4185A), using primer 5′-GAATCCATTTCTTCTTGCAAAGCTGTTTTGGCTCC-3′. To obtain enzymatic dead (ED) mutant PP5c, a histidine at position 304 was mutated to alanine (H304A) via two-step-PCR using the following primers: 1st reaction: 5′(wildtype): AAAAGAATTCACAGCATGACCATTGAGGAT, 3′(mutant): GTCTGTCTCGGCGTTGCCT; 2nd reaction: 5′(product of 1st PCR reaction), 3′(wildtype): TTTTCTCGAGCATCATTCCTAGCTGCAG.

GST-pulldown assays were conducted as described previously[32]. Briefly, a specific purified protein was incubated with a GST-fusion protein immobilized on glutathione sepharose beads at 4 °C for 1.5 h. Beads were then washed in high and low salt buffer three to four times. Samples were collected from each washing step including a sample of the beads. Analysis was performed by SDS-PAGE and western blot. Mostly, anti-PP5 antibody (target, N-terminal of human PP5; Cell Signaling, #2289; polyclonal, rabbit; 1:2000) or anti-PP5c (catalytic subunit) antibody (target, rat PP5 amino acids 36–238; 3/PP5; BD Biosciences, 611021; monoclonal, mouse; 1:2000) was used for detection, sometimes also anti-N2Bus (titin) antibody (custom-made by Eurogentec; affinity-purified polyclonal, rabbit; 1:500)[15,34] or anti-Hsp90(pan) antibody (target, peptide surrounding Asn300 of human Hsp90; C45G5; Cell Signaling, #4877 S; monoclonal, rabbit; 1:1000). A given interaction test was performed at least three times. Uncropped images of gels and western blots are shown in Supplementary Fig. 5 and Supplementary Fig. 6.

**Co-immunoprecipitation assay**. Prior to transfection, $1 \times 10^5$ HEK cells were seeded in 24-well plates and incubated at 37 °C and 5% $CO_2$ for 24 h. Cells were transfected (calcium phosphate method) using HA-tagged PP5 (3 µg DNA/well) and myc-tagged N2Bus (2 µg DNA/well). As DNA vector backbones we used pcDNA 3.1 (for PP5) and pCMV-Myc (for N2Bus). Twenty 4 h after transfection cells were rinsed twice with phosphate-buffered saline (PBS) and lysed by addition of 60 µl CellLytic M Reagent (Sigma-Aldrich) per well. PP5-HA was coupled to 20 µl anti-HA agarose beads in spin columns (Pierce HA tag IP/CoIP Kit, Thermo-Fisher Scientific) at 4 °C and 4 rpm overnight. Beads were spun down briefly and N2Bus-myc containing whole-cell lysate was added for 6 h at 4 °C and 4 rpm. After 10 s of pulse centrifugation the flow-through was discarded, beads were washed with PBS (3x) and bound protein eluted by addition of 100 µl non-reducing sample buffer at 95 °C for 5 min. Fractions were loaded on 10% SDS-PAGE gels at equal proportions; however, the eluate was additionally loaded at 7.5-fold excess, to reproducibly detect interaction). For western blotting, polyvinylidene difluoride (PVDF) membrane was blocked using 3% bovine serum albumin (BSA) at room temperature for 1 h. Primary antibodies were incubated at 4 °C overnight, secondary antibody at room temperature for 1 h. Signal detection of HRP conjugates was enhanced by ECL Western Blotting Detection kit (GE Healthcare). The following antibodies were used: anti-HA (target, HA peptide sequence (amino acids 76–111) of ×47 hemagglutinin; 3F10; Roche, #11867423001; monoclonal, rat; 1:4000) or anti-myc (target, aa 410–419 of human c-Myc; 9E10; Abcam, #ab32; monoclonal, mouse; 1:2000) as primary antibodies and anti-rat HRP (Dianova, #112-035-003; polyclonal, goat; 1:15,000) or anti-mouse HRP (Dianova, #115-035-003; polyclonal, goat; 1:10,000) as secondary antibodies.

**Immunofluorescence labeling of stretched single myofibrils**. Single myofibrils were isolated from human donor hearts (left ventricular tissue) and stretched in a custom-made setup, as described[32]. Briefly, single myofibrils were glued under an inverted microscope (Axiovert 135; Zeiss, Jena) to the tips of two microneedles, one connected to a rigid post, the other to a micromanipulator. Samples were rinsed in relaxing buffer (1 mmol/l of free Mg, 100 mmol/l KCl, 2 mmol/l EGTA, 4 mmol/l Mg-ATP, and 10 mmol/l imidazole, pH 7.0) at room temperature. Myofibrils were stretched and incubated for 20 min with recombinantly expressed catalytic subunit of PP5 (PP5c, 12 U/µl; activity measured using SensoLyte pNPP Protein Phosphatase Assay Kit (Anaspec)). Other myofibrils were pretreated for 30 min with PKA (recombinant catalytic subunit, 1 U/µl, Biaffin) before the incubation with PP5c. After 3 washing steps using relaxing buffer, bound PP5c was visualized by indirect immunofluorescence using primary anti-PP5c antibody (target, rat PP5 amino acids 36–238; 3/PP5; BD Biosciences, 611021; monoclonal, mouse; 1:50) and secondary Cy3-conjugated IgG (target, gamma immunoglobins heavy and light chains; Invitrogen, #A10520; polyclonal, goat; 1:400).

**Healthy and failing heart muscle tissues**. Left ventricular (LV) samples of non-failing human hearts were from brain-dead donors with normal LV function. Failing heart samples were from patients with dilated cardiomyopathy (DCM) who underwent heart transplantation due to severe systolic dysfunction (NYHA class III or IV). Sample collection was done in full accordance with Australian National Health Medical Research guidelines and approved by the Human Research Ethics Committee of the University of Sydney (HREC approval: 2012/2814). Informed consent was obtained from the next of kin of all organ donors. It is a requirement of the University of Sydney Human Research Ethics Committee that the consents not only must be sighted before the heart can be included in the Sydney Heart Bank[54], but any identifying details be redacted. Healthy dog heart tissue was obtained from adult mongrels (8–12 years) and failing heart samples from adult mongrel dogs made hypertensive by bilateral renal wrapping[55]. All dog samples were collected at the Mayo Clinic (Rochester, Minnesota, USA) in full accordance

with the institutional guidelines and approved by the Mayo Clinic Ethics Committee. Fetal (E18), newborn (P1) and adult heart tissue was obtained from female Wistar-Kyoto rats, following the guidelines and under the approval of the Animal Care and Use Committee at Ruhr University Bochum, Germany.

**PP5-overexpressing transgenic mouse hearts**. PP5 TG mice (CD1 strain) with cardiomyocyte-specific overexpression of the phosphatase under the α-myosin heavy-chain promoter were generated as described previously[25]. The transgene consisted of the α-myosin heavy-chain promoter, the entire protein coding region for rat PP5 (plus 483 base pairs of 3′ untranslated sequence), and the SV40 polyadenylation signal sequence. The transgene, isolated from the parent plasmid, was microinjected in fertilized mouse eggs. Mice positive for the transgene were identified via Southern blot and PCR of tail genomic DNA. In total, we studied the hearts of four TG and four litter-matched WT male mice aged 5–6 months, using a TG line with 7–8-fold overexpression of the phosphatase. These PP5 TG hearts showed a slightly compromised contractility in a previous study that provided phenotypic characterization of the model[25]. Hearts from TG and WT animals were collected at University of Muenster in full accordance with the institutional guidelines. Approval for the study was granted by the State Office for Nature, Environment and Consumer Protection North Rhine-Westphalia (LANUV NRW; reference number 8.87-50.10.36.09.006). Samples were prepared for immediate mechanical measurements of isolated cardiomyocytes, immunofluorescence, and immunoelectron microscopy, or deep-frozen for later biochemical analysis.

**FHL-1 knockout mouse hearts**. A mouse model deficient in FHL-1 (kind gift from Dr. Ju Chen, University of California-San Diego, La Jolla, CA, USA) was generated previously[29]. From this model we obtained adult heart tissue (6–8-month-old male mice) and analyzed three KO and three litter-matched WT hearts. All animal procedures were in full compliance with the guidelines approved by the UCSD Animal Care and Use Committee. The committee approved the procedure of heart extraction from the deceased animals for later biochemical and histochemical analysis.

**Neonatal rat cardiomyocyte cultures**. NRVM were isolated and prepared from 1-day-old to 2-day-old Wistar-Kyoto rat hearts[56], following the guidelines and under the approval of the Animal Care and Use Committee at Ruhr University Bochum. Cells were dispersed by incubation in pancreatin solution containing 8 g NaCl, 2 g D-glucose, 200 mg KCl, 57.5 mg $NaH_2PO_4$-$H_2O$, 20 mg phenol red, 1 g Na-bicarbonate, supplemented with 20 mg pancreatin (Sigma-Aldrich) and 150 U collagenase (Roche) per 100 ml at 37 °C. After several digestion steps, the pellet was suspended in Dulbecco's modified Eagle medium with 10% fetal calf serum (FCS) and antibiotics (10 mg/ml penicillin/streptomycin; Gibco). Pre-plating for 1 h removed unattached cells and fibroblasts and increased the content of cardiomyocytes. The cell suspension was then plated at a density of $1 \times 10^6$ cells/mm[2] and cultured at 37 °C and 5% $CO_2$. After 24 h, medium was replaced by fresh culture medium. Following another 2 h, some cells were treated with arachidonic acid (aa; Sigma-Aldrich; 200 µM, 2 h) or okadaic acid (oa; Sigma-Aldrich; 10 nM, 1 h), while others were left untreated ('control'). Additional cells were treated with inhibitor of Raf1 (cRaf1-I GW5074; Enzo Lifesciences; 20 µM; 1 h) or, to stimulate the MAPK pathway, with angiotensin-II (AngII; Sigma-Aldrich; 1 µM; 24 h), endothelin-1 (ET-1; Sigma-Aldrich; 10 nM; 24 h), or a combination of those. Cells were rinsed with warm PBS and subsequently harvested in 1.5 ml fresh PBS. After centrifugation, pellets were resolved in 40 µl modified Laemmli buffer (8 M urea; 2 M thiourea; 3 % SDS; 50 mM Tris-HCl (pH 6.8); 10% glycerol; 75 mM DTT; Serva Blue) and analyzed by SDS-PAGE and western blot for expression of PP5, ERK1/2, Phospho-ERK1/2, and GAPDH (loading control), using the following antibodies: anti-PP5 (target, N-terminal of human PP5; Cell Signaling, #2289; polyclonal, rabbit; 1:2000), anti-ERK1/2 (target, peptide corresponding to a sequence in the C-terminus of rat p44 MAP Kinase; Cell Signaling, #9102; polyclonal, rabbit; 1:1000), anti-P-ERK1/2 (target, phosphopeptide corresponding to residues surrounding Thr202/Tyr204 of ERK/MAPK; Biaffin, #AB-pERK-100; polyclonal, rabbit; 1:1000), and anti-GAPDH (target, full-length protein corresponding to human GAPDH; Abcam, #ab9484; monoclonal, mouse; 1:2000). P-ERK1/2 expression was normalized to ERK1/2 expression, indexed to GAPDH. Cells were prepared for indirect immunofluorescence and stained using antibodies against PP5 (target, N-terminal of human PP5; Cell Signaling, #2289; polyclonal, rabbit; 1:50), Z-disk marker α-actinin (target, rabbit skeletal α-actinin; EA-53; Sigma-Aldrich, #A7811; monoclonal, mouse; 1:300), and with Hoechst (ThermoFisher Scientific) to visualize the nuclei. Confocal laser scanning microscopy was done using the Nikon Eclipse Ti microscope.

**Adult rat cardiomyocyte cultures**. ARC were prepared as described[32]. Briefly, Wistar-Kyoto rats were euthanized with isoflurane, following the guidelines and under the approval of the local Animal Care and Use Committee. The abdomen was opened and ice-cold perfusion buffer (PB) with heparin was injected. The heart was excised and mounted on the cannula of a Langendorff perfusion system. Enzymatic digestion and rinsing were performed with different buffers in the cell isolation process, as described[32]. Cells were incubated in medium supplemented with 4% FCS, ITS (insulin/transferrin/selenium) and butanedione monoxime (BDM; 10 mM). Shortly after plating, this medium was exchanged for serum-free

medium (M199, Invitrogen; +ITS, +10 mM BDM). Ventricular cardiomyocytes were used for experiments from day 2 onwards. Cells were treated with arachidonic acid (aa, 200 µM; 2 h), or okadaic acid (oa, 10 nM; 1 h), while other cells were left untreated ('control'). Cells were then harvested as described above for NRVM. Total titin phosphorylation was detected on titin gels stained with the ProQ Diamond (phosphoprotein) vs. Sypro Ruby (total protein) system. Other cells were immunostained using antibodies against PP5 (target, N-terminal of human PP5; Cell Signaling, #2289; polyclonal, rabbit; 1:50) and α-actinin, (target, rabbit skeletal α-actinin; EA-53; Sigma-Aldrich, #A7811; monoclonal, mouse; 1:300) or phospho-N2Bus^S3991 (custom-made by Eurogentec; affinity-purified polyclonal, rabbit; 1:400)[15,34] and α-actinin, and indirect immunofluorescence was again performed using confocal laser scanning microscopy. On those samples, we counted the proportion of cells showing a clear striation pattern for either PP5 or titin P-S3991, relative to the total number of cells. Counting was done by a person blinded to the identity of the cells.

**Immunofluorescence microscopy**. Cultured cardiomyocytes or heart tissue sections were fixed in 4% paraformaldehyde (PFA) for 20 min, rinsed with PBS, and permeabilized with 0.5% triton X-100 for 5 min. After repeated washes with PBS, samples were blocked by 2% BSA for 20 min and subsequently stained with primary antibodies for 1 h at room temperature as follows: anti-α-actinin (target, rabbit skeletal α-actinin; EA-53; Sigma-Aldrich, #A7811; monoclonal, mouse; 1:300), anti-PP5 (target, N-terminal of human PP5; Cell Signaling, #2289; polyclonal, rabbit; 1:50), anti-PEVK-titin (custom-made by Eurogentec; affinity-purified polyclonal, rabbit; 1:40)[15,34], anti-phospho-N2Bus of titin (P-S3991; custom-made by Eurogentec; affinity-purified polyclonal, rabbit; 1:400)[15,34], anti-FHL-1 (target, immunogen corresponding to amino acids 23–121 of human FHL-1; Abcam; #ab58067; monoclonal, mouse; 1:250), anti-Hsp90 (target, peptide surrounding Asn300 of human Hsp90; C45G5; Cell Signaling, #4877S; monoclonal, rabbit; 1:20) and anti-phospho-Z/I-region of titin (P-S2080, custom-made by Eurogentec; affinity-purified polyclonal, rabbit; 1:100). Secondary antibodies for immunofluorescence were incubated for 1 h at room temperature at the following concentrations: Cy3-conjugated IgG (target, gamma immunoglobins heavy and light chains; Invitrogen, #A10520; polyclonal, goat), 1:300, and FITC-conjugated IgG (target, mouse IgG whole molecule; Rockland, #210–1204; polyclonal, goat), 1:200. In control experiments, secondary antibody alone was used, but did not reveal any immunofluorescence signals above background. Immunostained samples were embedded in Mowiol (Sigma-Aldrich) and analyzed by confocal laser scanning microscopy (Nikon Eclipse Ti microscope). Immunofluorescence imaging was processed similarly in the experimental and control groups.

**Immunoelectron microscopy**. PP5 TG, FHL-1 KO mouse hearts, and matching WT hearts were fixed in 4% PFA, 15% saturated picric acid in 0.1 M phosphate buffer (pH 7.4) overnight at 4 °C and cut into longitudinal sections on a vibratome (Leica VT 1000 S). Thin sections were blocked in 20% normal goat serum (NGS) and incubated with primary antibodies in PBS supplemented with 5% NGS overnight at 4 °C. The following antibodies were used: anti-PP5 (target, N-terminal of human PP5; Cell Signaling, #2289; polyclonal, rabbit; 1:50), anti-P-S3991 of N2Bus-titin (custom-made by Eurogentec; affinity-purified polyclonal, rabbit; 1:400)[15,34], or anti-P-S2080 of Z/I-region of titin (P-S2080, custom-made by Eurogentec; affinity-purified polyclonal, rabbit; 1:100). Sections were rinsed and incubated with 1.4 nm gold-coupled secondary antibodies (Nanoprobes, Stony Brook, NY, USA) overnight at 4 °C. After extensive washing, sections were post-fixed in 1% glutaraldehyde for 10 min and then reacted with HQ Silver kit (Nanoprobes). $OsO_4$ and uranyl acetate were used for counterstaining. After dehydration via ascending ethanol series, the samples were embedded in Durcupan resin (Fluka, Switzerland). Ultrathin sections (Ultracut S, Leica, Germany) were prepared and analyzed using a Zeiss LEO 910 electron microscope. For quantification of results, the nanogold particles were counted in 50 µm[2]-sized regions-of-interest, distinguishing the particles bound to titin in the sarcomeric I-band ('On I-band') vs. those located elsewhere in the cardiomyocyte ('Not on I-band'). The counting person was blinded to genotype/phenotype.

**SDS-PAGE and immunoblotting**. Deep-frozen cardiac tissues (or cardiomyocyte cultures) were homogenized in modified Laemmli buffer, stored on ice for 30 min and subsequently boiled for 3 min at 97 °C. Protein concentration was determined spectrophotometrically using the method of Neuhoff[57]. Standard SDS-PAGE was employed for proteins of 'normal' size and agarose-strengthened 2% SDS-PAGE for titin analysis[58]. Western blotting was performed as described[59]. Antibodies to the following proteins were used: PP5 (target, N-terminal of human PP5; Cell Signaling, #2289; polyclonal, rabbit; 1:2000), PP5c (target, rat PP5 amino acids 36–238; 3/PP5; BD Biosciences, 611021; monoclonal, mouse; 1:2000), GAPDH (target, full-length protein corresponding to human GAPDH; Abcam, #ab9484; monoclonal, mouse; 1:2000), PP1α (target, peptide corresponding to the N-terminal sequence of human PP1α; Cell Signaling, #2582; polyclonal, rabbit; 1:1000), PP2a (α + β isoform; target, peptide corresponding to amino acids at the C-terminus of human PP2A catalytic subunit; Cell Signaling, #2038; polyclonal, rabbit; 1:1000), phospho-Raf1 (Ser338; target, phosphopeptide corresponding to residues 333–345 of human Raf-1; Merck, 05–538; monoclonal, mouse; 1:1000),

Hsp90 (target, peptide surrounding Asn300 of human Hsp90; C45G5; Cell Signaling, #4877S; monoclonal, rabbit; 1:1000), ERK1/2 (target, peptide corresponding to a sequence in the C-terminus of rat p44 MAP kinase; Cell Signaling, #9102; polyclonal, rabbit; 1:1000) and phospho-ERK1/2 (target, phosphopeptide corresponding to residues surrounding Thr$^{202}$/Tyr$^{204}$ of ERK/MAPK; Biaffin, #AB-pERK-100; polyclonal, rabbit; 1:1000).

**Quantification of titin phosphorylation**. By $^{32}$P-autoradiography we measured phosphorylation of recombinant human N2Bus[15]. N2Bus fragment was incubated with PKG (4.7 U/µl + 10 mM cGMP; kindly provided by Dr. Elke Butt, Würzburg, Germany) or PKA catalytic subunit (1 U/µl; Biaffin) in presence of [γ$^{32}$P]ATP (250 µCi/µM; Hartmann Analytic, Braunschweig, Germany) for 30 min at 35 °C. Some samples were dephosphorylated overnight at 35 °C with recombinant full-length PP5 or catalytic subunit of PP5 (PP5c) (12 U/µl). Samples were loaded on 7.5% SDS-PAGE gels, which were dried and exposed to autoradiographic film (Fujifilm BAS-1800 II) overnight at room temperature. Signals were analyzed using Multi Gauge V3.2 software.

The ProQ Diamond/Sypro Ruby dual staining system (Molecular Probes) was used to measure titin phosphorylation in recombinant N2Bus, rat cardiomyocytes, or adult mouse heart tissue. Recombinant human N2Bus was phosphorylated with PKG (4.7 U/µl, plus 10 mM cGMP), PKA catalytic subunit (1 U/µl), or ERK2 (20.5 U/ml; ThermoFischer Scientific) in presence of ATP for 30 min at 35 °C, and was dephosphorylated overnight at 35 °C using recombinant full-length PP5 or catalytic subunit of PP5 (PP5c) (12 U/µl). A separate batch of recombinant N2Bus was also treated (following phosphorylation) with enzymatic dead PP5c (ED H304A mutant) overnight at 35 °C. Proteins were separated on SDS-PAGE gels stained with ProQ Diamond for 1 h and subsequently with Sypro Ruby overnight. Proteins were visualized using the LAS-4000 image reader (Fujifilm). Protein bands were analyzed densitometrically and phospho-signals on ProQ Diamond-stained gels were normalized to corresponding total protein signals on Sypro Ruby-stained gels.

By western blotting, we measured site-specific titin phosphorylation in heart tissue, as described[15,34]. Following separation of titin on loose SDS-PAGE gels, immunoblotting was performed using custom-made, affinity-purified, phospho-specific antibodies against phosphoserines in titin (for details, see ref. [15]). The following anti-mouse titin antibodies (affinity-purified, polyclonal, rabbit), generated by Eurogentec (Belgium), were used (amino acid positions according to UniProtKB identifier, A2ASS6): anti-P-S2080 (Z/I-junctional region; 1:400); anti-P-S3991 (1:400); -P-S4043 (1:400); -P-S4080 (1:400) (all N2Bus); and anti-P12742 (1:400) (PEVK domain)[15,34]. For each phosphosite-specific anti-mouse antibody, a corresponding pan-titin antibody recognizing both the phosphorylated and the non-phosphorylated site was used separately on western blots. For quantification of phosphorylation, phospho-titin signals were normalized to signals obtained with the corresponding pan-titin antibodies. Moreover, phospho-specific anti-human titin (UniProtKB identifier, Q8WZ42) antibodies (affinity-purified, polyclonal, rabbit) were used that have been generated by Eurogentec against P-S4010 (1:400), P-S4099 (1:400) (both N2Bus), P-S11878 (1:400), and P-12022 (1:400) (both PEVK)[15,34]. Another antibody (affinity-purified, polyclonal, rabbit) was custom-made by Immunoglobe (Himmelstadt, Germany) against P-S4185 in human N2Bus (1:500)[34]; this site is present in human titin, but is not cross-species conserved[13]. Signals obtained with the anti-human antibodies were indexed to signals on (coomassie-stained) PVDF blotting membrane, to adjust for differences in protein load. In some experiments, endogenous titin in human heart samples was first dephosphorylated using alkaline phosphatase (AP, 10 U/µl; New England Biolabs) before back-phosphorylation with PKG, followed by dephosphorylation with PP5 or PP5c. Uncropped images of gels, autoradiogrammes, and western blots are shown in Supplementary Fig. 5 and Supplementary Fig. 6.

**Force measurements on isolated skinned cardiomyocytes**. Force measurements were performed on single, isolated cardiomyocytes, as described[15]. Cardiomyocytes were isolated from the hearts of PP5 TG ($n = 3$) and litter-matched WT ($n = 3$) mice. Cells were skinned in relaxing buffer by using 0.5% Triton X-100 and placed under a Zeiss Axiovert 135 inverted microscope (×40 objective). Single cardiomyocytes were suspended between a force transducer and a motor, which are part of a 'Permeabilized Myocyte Test System' (1600A; Aurora Scientific, Aurora, Canada). Experiments were performed in relaxing solution at room temperature. SL was set to slack (1.8–1.9 µm SL) and then increased step-wise to 2.4 µm while the force was recorded. 'Passive' force values were normalized to myocyte cross-sectional area calculated from the diameter of the cells, assuming a circular shape. In separate sets of experiments, PP5 WT or TG cardiomyocytes in relaxing buffer were incubated with one of the following enzymes: catalytic subunit of PP5 (PP5c, 12 U/µl, used at 0.6 µg/µl), ERK2 (20.5 U/ml), PKG (0.1 U/ml) in presence of activator cGMP (10 mM), or PKA (catalytic subunit) (100 U/ml). Sometimes, the PP5 was added after prior incubation with a kinase. Incubation time per enzyme was 40 min at 20 °C. On passive tension vs. SL plots, mean data points were fit by third-order polynomials.

**Statistical analysis**. Values are given as mean ± s.e.m. of n observations in each group; biological replicates are indicated. Means were compared using Bonferroni adjusted t-test or two-tailed Student's t-test, if data passed the normal distribution test. For multiple comparisons, the Holm-Sidak method was applied. Data needed

to pass the equal variance test. Three levels of significance were defined: *$p < 0.5$; **$p < 0.01$; and ***$p < 0.001$. Analyses were performed in GraphPad Prism 5 or SigmaPlot 12.5.

**Data availability**. All relevant data are available from the authors upon reasonable request.

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

## Acknowledgements

We thank Sebastian Kötter for providing recombinant mutant N2Bus fragment, Patrick Lang for help with single-myofibril mechanics, Wolfgang Obermann for providing Hsp90 vectors, Elke Butt for providing cGMP-activated PKG, Ju Chen for the kind gift of the FHL-1 KO mouse model, and Gabriele Reimus for help with figure preparation. This work was supported by grants from the German Research Foundation (SFB1002, TPA08 and HA 7512/2-1).

## Author contributions

J.K. carried out most of the experiments and wrote a manuscript draft. A.U. performed immunofluorescence and immunoelectron microscopy. L.B. performed binding assays in cell culture. N.H. carried out mechanical measurements on cardiomyocytes. M.v.F.-S. performed protein gel electrophoresis and western blotting. C.G.d.R. and M.M.R. provided tissue from failing hearts. F.S., U.G., and P.B. generated and provided genetic mouse models and revised the manuscript. W.A.L. designed and supervised the study, analyzed data, wrote the manuscript, and provided funding.

## Additional information

**Competing interests:** The authors declare no competing financial interests.

