## [Peer review file · Nature Communications]

Reviewers' comments:

Reviewer #1

Remarks to the Author:

PP5 is a ser/thr protein phosphatase (PP5/PPP5C) that is ubiquitously expressed. PP5 expression has been observed in cardiomyocytes; however the role of PP5 in hearts has not been explored. In this potentially very interesting study, the authors present fairly convincing data indicating PP5 dephosphorylates Titin, reversing titin-based passive tension in the heart. They present convincing data showing that PP5 is a novel binding partner for Titin N2Bus. PP5 is known to act on Raf1. However, to my knowledge this is the first report to provide data that PP5-mediated dephosphorylation of Raf1 affects N2Bus-associated mechanosensor complex. This is also an important finding.

Their observation that PP5 protein levels are elevated in both human and dog failing hearts is novel and may prove to be a very important finding. Studies with transgenic mice that over express PP5 specifically in cardiomyocytes appear to validate observations made in vitro. In general this is an impressive study that may have a large and lasting impact on the field. I have a few concerns that are listed below.

Concerns.

1) In figure 1 B and E and in figure 6 A, B and C the input bands are cropped to close and some of the data is lost due to cropping (i.e. there are black squares where there should be bands). These images should be redone taking care to make sure the cropping and the size of the bands are not accidentally enlarged as the bands are arranged to construct the figures.

2) In the discussion the authors suggest that spermine may act to inhibit PP5C and may thus be useful as a therapeutic option for HFpEF patients. Does spermine have any effect on their transgenic mice that over express PP5? This would be very interesting data that would greatly enhance the interest of this manuscript.

3) In Figure 6 they show a novel association between PP5 and PP1 using a GST-pull down experiment, and they indicate in the text that PP5 is known to form a heterodimer with PP2A. While PP5 has been reported to form a heterodimer with PP2A, it does not bind the catalytic subunit as implied in the discussion. Rather PP5 has been reported to associate with a B-regulatory/targeting subunit that also binds PP2A. This part of the discussion is misleading and should be corrected.

3a) More importantly, they report a novel association between PP5 and PP1 based on data from a single GST-pull down experiment. The direct interaction between PP5 and PP1C has not been reported previously, and as stated it appears this interaction is a direct interaction between PP5 and PP1C. Based on the crystal structures indicating that the catalytic subunits of both are highly conserved it is difficult to envision how this interaction can occur without the catalytic subunits of PP1, PP2A or PP5 forming homo or hetero dimers with each other. Therefore it seems more likely that the observed interaction requires an additional linker protein. PP1 has been reported to associate with >100 targeting/regulatory proteins. Do any of these show up in the GST-pull down. How clean is the pull down? A coomassie or silver stain of the GST-pull down should be provided as supplemental data to demonstrate the specific nature of the interaction.

If PP5 forms a direct interaction with PP1C in biological setting in the heart, this is an important finding, however such a bold claim should not be made without providing much more substantial data demonstrating the interaction is real and meaningful. As is, the speculation that PP5 recruits PP1 to a particular location distracts from a very interesting story about the role of PP5C in N2Bus-associated mechanosensor complex.

Minor concerns.

- 1) For the majority of the manuscript the old nomenclature is used, often without reference to the new names. It would be helpful to younger readers if the new nomenclature is introduced along with the old names the first time the protein is introduced. (e.g. PP5/PPP5C; ERK2/ MAPK1 etc)
- 2) In figure 6 a (bottom left) I think the PP5 second from the bottom should be labeled PP5C. If not the kDa 35 must be wrong.
- 3) In figure 8; okadaic acid is misspelled (Ocadaic in figure)

Reviewer #2

Remarks to the Author:

This manuscript deals with the role of PP5 in the heart and proposes its importance in the regulation of titin stiffness and in a mechanosensing complex in the sarcomere. Phosphatases other than calcineurin are currently underexplored in the heart, so this is a welcome contribution to the field. The authors show that the levels of PP5 expression in cardiomyocytes are dependent on developmental stage and are elevated in failing hearts. Cell culture experiments indicate that activated PP5 has an increased half life, potentially due to being sequestered to the titin N2Bus region. Overexpression of PP5 in transgenic mice or addition of recombinant PP5c leads to increased passive stiffness in myofibrils. In conclusion the authors propose the existence of a multiprotein complex at the titin N2Bus site that is involved in mechanosensing and to which PP5 is recruited in heart failure, thus contributing to the dephosphorylation of titin there and increasing passive stiffness.

Most of the data are convincing and my main comments are on presentation and technical issues.

1. Why opt for this extreme cropping of bands in most of the Western blot Figures and why are samples hardly ever run consecutively? This approach does not help in assessing whether the smudge that is shown for bound T+ in Figure 1b is real enough to justify narrowing down the binding site to just six amino acids. There are also weird horizontal black bars in some of the bands (3rd and 4th input from the top in 1e, bound in the top row of 6b and the 2nd input from the top in 6c)? What are these? Also the shadow shown for N2Bus-myc shown in Figure 1c is not convincing in its cropped isolation. Would addition of arachidonic acid to the HEK cultures improve the efficiency of the immunoprecipitation assay?
2. Cardiomyocytes themselves only express minimal amounts of the beta-cytoplasmic actin isoform (see e.g. Lin & Redies, 2012), therefore this is not a meaningful loading control since it will just assess for the amounts of contaminating non myocytes in the cultures/tissue (Figure 2b, Figure 3f, Figure 4a).
3. What are the fluorescent streaks running perpendicular to the alpha actinin signal in Figure 2d and f? Bleedthrough? The massive fluorescence signal for PP5 in Figure 2d is also unexpected given the expression data from Figure 2a. Could this all be due to autofluorescence from mitochondria?
4. Any explanation why full length recombinant PP5 would be active (Figure 3a, 3d)? Should it not be autoinhibited?
5. Show also original data for P-S4010 in the human and dog samples and not just the quantification (Figure 3e).
6. What about PP5 expression levels and subcellular localization in FHL-1 knockout mice?

Response to Reviewers (NCOMMS-17-03357A)

Response to Reviewer 1

“PP5 is a ser/thr protein phosphatase (PP5/PPP5C) that is ubiquitously expressed. PP5 expression has been observed in cardiomyocytes; however the role of PP5 in hearts has not been explored. In this potentially very interesting study, the authors present fairly convincing data indicating PP5 dephosphorylates Titin, reversing titin-based passive tension in the heart. They present convincing data showing that PP5 is a novel binding partner for Titin N2Bus. PP5 is known to act on Raf1. However, to my knowledge this is the first report to provide data that PP5-mediated dephosphorylation of Raf1 affects N2Bus-associated mechanosensor complex. This is also an important finding.

Their observation that PP5 protein levels are elevated in both human and dog failing hearts is novel and may prove to be a very important finding. Studies with transgenic mice that over express PP5 specifically in cardiomyocytes appear to validate observations made in vitro. In general this is an impressive study that may have a large and lasting impact on the field. I have a few concerns that are listed below.”

Thank you very much for your insightful comments and suggestions. Below is a detailed response to your critique.

“Concerns.

1) In figure 1 B and E and in figure 6 A, B and C the input bands are cropped to close and some of the data is lost due to cropping (i.e. there are black squares where there should be bands). These images should be redone taking care to make sure the cropping and the sized of the bands are not accidentally enlarged as the bands are arranged to construct the figures.”

Please accept our apologies for this sloppiness. In this revision, we have taken care to mend all technical issues related to the cropping of bands in the Western blot/gel figures. As you can see, the extreme cropping is now avoided throughout the manuscript. Moreover, for all gels and blots shown in the main and supplementary figures, we provide the full scans in the supplementary dataset, listed under Supplementary Figures 5 and 6. Occasionally it became necessary to re-run a gel or blot for this revision. In the few remaining cases, in which non-contiguous bands are grouped, they were nevertheless taken from the same gel, as can be seen on the full scans.

“2) In the discussion the authors suggest that spermine may act to inhibit PP5C and may thus be useful as a therapeutic option for HFpEF patients. Does spermine have any effect on their transgenic mice that over express PP5? This would be very interesting data that would greatly enhance the interest of this manuscript.”

Your suggestion to test whether spermine has an effect on PP5-overexpressing TG mice arose from our discussion of the possibility that spermidine could also inhibit PP5, a point mentioned in light of our recent work showing that spermidine treatment of aged mice or hypertensive rats may affect titin stiffness via alterations in titin phosphorylation (our Ref. 53). Your suggestion could be addressed in future work, but we hope you agree that such an experiment would distract from the principal message of our present manuscript, which is about basic PP5 properties and functions in cardiomyocytes. Moreover, spermine (and any related polyamine) is a pleiotropic drug that has multiple effects on molecular, cellular, organ, and organismal functions, unrelated to its relationship with PP5 (e.g., it is an autophagy inducer and it also promotes the cGMP-PKG pathway, which phosphorylates titin, among others), and any possible effect measured *in vivo* would be difficult to interpret straightforwardly. In our opinion it would require extensive biochemical work to prove a direct link between polyamine administration and PP5 functional changes *in vivo*. Another point is that the cardiac specific PP5-overexpressing TG mouse model used by us has a modest systolic heart phenotype (our Ref. 25), but there was no beneficial effect of

spermidine on systolic function in the aged mice or hypertrophic rats; instead, there was a beneficial effect on diastolic function, e.g., de-stiffening of the cardiac walls (our Ref. 53). Therefore, an effect of spermine on cardiac function in the PP5 TG hearts may not be obvious. Despite these considerations, we believe that pointing out the possible link between spermidine/spermine, PP5, and titin phosphorylation, could nevertheless be useful to the interested reader, and we have kept this speculative part of the Discussion in the current manuscript.

“3) In Figure 6 they show a novel association between PP5 and PP1 using a GST-pull down experiment, and they indicate in the text that PP5 is known to form a heterodimer with PP2A. While PP5 has been reported to form a heterodimer with PP2A, it does not bind the catalytic subunit as implied in the discussion. Rather PP5 has been reported to associate with a B-regulatory/targeting subunit that also binds PP2A. This part of the discussion is misleading and should be corrected.”

We agree that the literature is divided about how the interaction between PP5 and PP2a is brought about, and that an association of both PP5 and PP2a with a B-regulatory/targeting subunit may be critical. We have therefore rephrased the text on p.15 (last paragraph), which now reads:

“Second, PP5 can associate with PP2a, either directly⁴⁸ or indirectly through a regulatory subunit that binds both phosphatases⁴⁹. PP2a is known to dephosphorylate multiple cardiomyocyte proteins^{38,50} and it has been used experimentally to dephosphorylate titin *in vitro*¹³.”

“3a) More importantly, they report a novel association between PP5 and PP1 based on data from a single GST-pull down experiment. The direct interaction between PP5 and PP1C has not been reported previously, and as stated it appears this interaction is a direct interaction between PP5 and PP1C. Based on the crystal structures indicating that the catalytic subunits of both are highly conserved it is difficult to envision how this interaction can occur without the catalytic subunits of PP1, PP2A or PP5 forming homo or hetero dimers with each other. Therefore it seems more likely that the observed interaction requires an additional linker protein. PP1 has been reported to associate with >100 targeting/regulatory proteins. Do any of these show up in the GST-pull down. How clean is the pull down? A coomassie or silver stain of the GST-pull down should be provided as supplemental data to demonstrate the specific nature of the interaction.

If PP5 forms a direct interaction with PP1C in biological setting in the heart, this is an important finding, however such a bold claim should not be made with out providing much more substantial data demonstrating the interaction is real and meaningful. As is, the speculation that PP5 recruits PP1 to a particular location distracts from a very interesting story about the role of PP5C in N2Bus-associated mechanosensor complex.”

We agree with the reviewer about this issue. Our initial response to this point was to perform exploratory immunostaining experiments on myocardial sections with the goal to detect PP1 at the known PP5-N2Bus interaction site in the middle of the elastic I-band titin region. However, these experiments did not reveal any hint of a localization of PP1 at elastic I-band titin in the sarcomere, unlike for PP5. The regular striation pattern in cardiomyocytes shown for PP1 in the literature may be due to its association with SERCA/phospholamban in the sarcoplasmic reticulum (Qian J, Vafiadaki E, Florea SM, Singh VP, Song W, Lam CK, Wang Y, Yuan Q, Pritchard TJ, Cai W, Haghghi K, Rodriguez P, Wang HS, Sanoudou D, Fan GC, Kranias EG. Small heat shock protein 20 interacts with protein phosphatase-1 and enhances sarcoplasmic reticulum calcium cycling. *Circ Res* 2011;108:1429-38). As it stands, then, the positive *in vitro* binding experiment suggestive of a PP1-PP5 interaction might not be indicative of the situation in cardiomyocytes and in a biological setting in the heart. In the absence of evidence for the co-localization, PP1 also does not appear to be part of the PP5-MAPK-FHL-N2Bus mechanosensor complex in cardiomyocytes. Therefore, and in order to not further distract “...from a very interesting

story about the role of PP5C in N2Bus-associated mechanosensor complex”, we have opted for removing our single GST-pulldown result probing the PP5-PP1 interaction (previously in Figure 6a). We have also removed PP1 from our model in Figure 8. We have only kept a sentence in the discussion/outlook, which briefly mentions PP1 (p. 15, bottom):

“Moreover, PP5 could be linked structurally or functionally to other important phosphatases in cardiomyocytes, such as PP1 and calcineurin^{2,38,51,52}, which is a possibility worth studying in the future.”

“Minor concerns.

1) For the majority of the manuscript the old nomenclature is used, often without reference to the new names. It would be helpful to younger readers if the new nomenclature is introduced along with the old names the first time the protein is introduced. (e.g. PP5/PPP5C; ERK2/ MAPK1 etc)”

Thank you, we have now provided the new names on first mentioning of the old names.

“2) In figure 6 a (bottom left) I think the PP5 second from the bottom should be labeled PP5C. If not the kDa 35 must be wrong.”

This was the panel showing the PP1-PP5 interaction, which has been removed.

“3) In figure 8; okadaic acid is misspelled (Ocadaic in figure)”

Thank you, we have corrected this misspell in the text and in Figure 8.

Response to Reviewer 2

“This manuscript deals with the role of PP5 in the heart and proposes its importance in the regulation of titin stiffness and in a mechanosensing complex in the sarcomere. Phosphatases other than calcineurin are currently underexplored in the heart, so this is a welcome contribution to the field. The authors show that the levels of PP5 expression in cardiomyocytes are dependent on developmental stage and are elevated in failing hearts. Cell culture experiments indicate that activated PP5 has an increased half life, potentially due to being sequestered to the titin N2Bus region. Overexpression of PP5 in transgenic mice or addition of recombinant PP5c leads to increased passive stiffness in myofibrils. In conclusion the authors propose the existence of a multiprotein complex at the titin N2Bus site that is involved in mechanosensing and to which PP5 is recruited in heart failure, thus contributing to the dephosphorylation of titin there and increasing passive stiffness.

Most of the data are convincing and my main comments are on presentation and technical issues.”

Thank you very much for your insightful comments and suggestions. Below is a detailed response to your critique.

“1. Why opt for this extreme cropping of bands in most of the Western blot Figures and why are samples hardly ever run consecutively? This approach does not help in assessing whether the smudge that is shown for bound T+ in Figure 1b is real enough to justify narrowing down the binding site to just six amino acids. There are also weird horizontal black bars in some of the bands (3rd and 4th input from the top in 1e, bound in the top row of 6b and the 2nd input from the top in 6c)? What are these? Also the shadow shown for N2Bus-myc shown in Figure 1c is not convincing in its cropped isolation.”

Please accept our apologies for this sloppiness. In this revision, we have taken care to mend all technical issues related to the cropping of bands in the Western blot/gel figures. As you can see, the extreme cropping is now avoided throughout the manuscript. Moreover, for all gels and blots shown in the main and supplementary figures, we provide the full scans in the supplementary dataset, listed under Supplementary Figures 5 and 6. Occasionally it became necessary to re-run a gel or blot for this revision. In the few remaining cases, in which non-contiguous bands are grouped, they were nevertheless taken from the same gel, as can be seen on the full scans.

“Would addition of arachidonic acid to the HEK cultures improve the efficiency of the immunoprecipitation assay?”

The addition of arachidonic acid did not significantly enhance the efficiency of the Co-IP assay.

“2. Cardiomyocytes themselves only express minimal amounts of the beta-cytoplasmic actin isoform (see e.g. Lin & Redies, 2012), therefore this is not a meaningful loading control since it will just assess for the amounts of contaminating non myocytes in the cultures/tissue (Figure 2b, Figure 3f, Figure 4a).”

In response, we have removed beta-actin as a loading control and replaced it by GAPDH throughout. New gels have been run, where necessary, and also the analysis has been re-done.

“3. What are the fluorescent streaks running perpendicular to the alpha actinin signal in Figure 2d and f? Bleedthrough? The massive fluorescence signal for PP5 in Figure 2d is also unexpected given the expression data from Figure 2a. Could this all be due to autofluorescence from mitochondria?”

The unexpected fluorescence streaks for alpha-actinin present in the former Figures 2d and 2f were not due to bleedthrough or autofluorescence of mitochondria, but most likely, to overexposure of the images. We have corrected this and now show less-exposed images without those streaks. We also added two sentences in the Methods section (p.22, second paragraph) mentioning that:

“In control experiments, secondary antibody alone was used, but did not reveal any immunofluorescence signals above background.” and

“Immunofluorescence imaging was processed similarly in the experimental and control groups.”

“4. Any explanation why full length recombinant PP5 would be active (Figure 3a, 3d)? Should it not be autoinhibited?”

We were surprised, too, that full-length PP5 was more active than expected. A sentence has been added which addresses this point (p.8, end of second paragraph):

“The relatively high phosphatase activity of full-length PP5 found in these experiments was unexpected but consistently observed; it might be due to partial cleavage (and thus, activation) of the phosphatase in vitro or incomplete autoinhibition.”

“5. Show also original data for P-S4010 in the human and dog samples and not just the quantification (Figure 3e).”

We apologize for not showing these before. The original data for P-S4010 are now included, along with new site-specific titin phosphorylation results for this set of 10 donor and 10 failing human hearts, using phospho-specific antibodies to two additional N2Bus sites and 2 (previously not shown) PEVK sites (Figure 3e). Results demonstrate that site-specific phosphorylation of N2Bus (the interactor and substrate of PP5) is reduced in heart failure, whereas PEVK phosphorylation is unaltered compared to healthy hearts. These findings underscore our conclusion that induction of PP5 in heart failure contributes to hypo-phosphorylation of titin specifically at N2Bus, which causes the pathologically increased myocardial stiffness frequently observed in various types of heart failure. The relevant text in Results and Discussion was updated. Please note that the dog heart results (formerly in Figure 3e, f) have now been placed in Supplementary Figure 2a, b.

“6. What about PP5 expression levels and subcellular localization in FHL-1 knockout mice?”

In response, we have now provided new immunofluorescence images in Figure 7b demonstrating the localization of PP5 in FHL-1 knockout and WT mouse heart tissue. On p.12, last para, we also included the following text:

“On immunofluorescently stained tissue sections of both WT and FHL-1 KO hearts, PP5 appeared mainly in the cytosolic space and sometimes at the sarcomeres; a difference in PP5-staining intensity and pattern was not consistently observed (Fig. 7b).”

REVIEWERS' COMMENTS:

Reviewer #1 (Remarks to the Author):

In the revision, the authors addressed all of my previous concerns. This interesting manuscript is ready for publication.

richard honkanen

Reviewer #2 (Remarks to the Author):

I am content with the revisions that were made.

RESPONSE TO REVIEWERS' COMMENTS (NCOMMS-17-03357A)

Reviewer #1 (Remarks to the Author):

In the revision, the authors addressed all of my previous concerns. This interesting manuscript is ready for publication.

Reviewer #2 (Remarks to the Author):

I am content with the revisions that were made.

We thank the reviewers for their constructive and insightful comments.